behaviour/ecology

deep-sea food web, scattering layer, deep-diving, optimal foraging, animal decision-making, *Grampus griseus*

**Author for correspondence:**
Fleur Visser
e-mail: f.visser@uva.nl

# Risso's dolphins perform spin dives to target deep-dwelling prey

Fleur Visser[1,2,3], Onno A. Keller[1,2,4], Machiel G. Oudejans[3], Douglas P. Nowacek[5,6], Annebelle C. M. Kok[3,7,8], Jef Huisman[1] and Elisabeth H. M. Sterck[4,9]

[1]Department of Freshwater and Marine Ecology, Institute for Biodiversity and Ecosystem Dynamics, University of Amsterdam, PO Box 94240, 1090 GE, Amsterdam, The Netherlands
[2]Department of Coastal Systems, NIOZ Royal Netherlands Institute for Sea Research, PO Box 59, 1790 AB, Den Burg, Texel, The Netherlands
[3]Kelp Marine Research, 1624 CJ, Hoorn, The Netherlands
[4]Department of Biology, Utrecht University, 3584 CH, Utrecht, The Netherlands
[5]Nicholas School of the Environment, Duke University Marine Laboratory, Beaufort, NC 28516, USA
[6]Pratt School of Engineering, Duke University, Durham, NC 27708, USA
[7]Institute of Biology, Leiden University, PO Box 9509, 2300 RA, Leiden, The Netherlands
[8]Scripps Institution of Oceanography, UCSD, La Jolla 92093–0205, USA
[9]Animal Science Department, Biomedical Primate Research Centre, 2288 GJ, Rijswijk, The Netherlands

FV, 0000-0003-1024-3244; OAK, 0000-0002-3443-1234; ACMK, 0000-0001-6619-0191

Foraging decisions of deep-diving cetaceans can provide fundamental insight into food web dynamics of the deep pelagic ocean. Cetacean optimal foraging entails a tight balance between oxygen-conserving dive strategies and access to deep-dwelling prey of sufficient energetic reward. Risso's dolphins (*Grampus griseus*) displayed a thus far unknown dive strategy, which we termed the spin dive. Dives started with intense stroking and right-sided lateral rotation. This remarkable behaviour resulted in a rapid descent. By tracking the fine-scale foraging behaviour of seven tagged individuals, matched with prey layer recordings, we tested the hypothesis that spin dives are foraging dives targeting deep-dwelling prey. Hunting depth traced the diel movement of the deep scattering layer, a dense aggregation of prey, that resides deep during the day and near-surface at night. Individuals shifted their foraging strategy from deep spin dives to shallow non-spin dives around dusk. Spin dives were significantly faster, steeper and deeper than non-spin dives, effectively minimizing transit time to bountiful mesopelagic prey, and were focused on periods when the migratory prey might be easier to catch. Hence, whereas Risso's

dolphins were mostly shallow, nocturnal foragers, their spin dives enabled extended and rewarding diurnal foraging on deep-dwelling prey.

# 1. Background

Understanding species interactions in the deep sea is one of the key challenges in the study of oceanic food webs [1]. However, due to the difficulty of observation of both predator and prey, knowledge on meso- and bathypelagic predator–prey systems remains virtually non-existent. As top predators, cetaceans are key drivers of oceanic food web structure [2]. During foraging, they face a trade-off between selective forces arising from oxygen uptake at the surface and prey capture at depth. Spatial separation between these two vital resources is specifically pronounced in deep-diving toothed whales, driving the evolution of energy-conserving dive strategies [3].

Deep dives are costly for air-breathing marine predators, as the increased temporal and energetic costs of travel, in combination with physiological restrictions, constrain effective foraging time at depth [4,5]. Minimizing cost of travel is thus essential to maintain optimal foraging [6] and deep-diving cetaceans have evolved specialized diving, oxygen-conserving and biosonar strategies to target and locate deep-dwelling prey [4,7–9].

The deep-diving Risso's dolphin (*Grampus griseus*) hunts cephalopods and fish using biosonar [10–13]. Individuals target different foraging zones and can actively switch between shallower and deeper dives. Their target foraging depth, around 50–600 m, is determined prior to dive-onset, using the information on prey layers and foraging performance obtained in previous dives [14–16]. Prior to their dives, Risso's dolphins may not only actively choose their target depth, but also optimize their dive and movement strategy. Here, we report on a unique dive type observed in Risso's dolphins. This dive starts with intense stroking at the surface, followed by a rapid lateral rotation onto the side, a subsequent turn downwards and rapid descent into deeper waters (electronic supplementary material, video S1). This 'spin dive' has not been described previously and its function remains untested.

We hypothesize that spin dives function to optimize Risso's dolphin foraging performance when targeting deep-dwelling prey in the pelagic environment. We tested this hypothesis by comparing the fine-scale foraging behaviour associated with spin dives and non-spin dives of Risso's dolphins in the Azores, equipped with suction cup-attached sound and movement recording tags [17]. These data were matched with *in situ* data on prey layer depth, obtained from echo sounders. By dual observation of deep-diving dolphins and their prey, we further investigate the foraging strategies employed by air-breathing marine predators and provide rare insight into predator–prey interactions of the deep pelagic ocean.

# 2. Methods

## 2.1. Data collection

Fieldwork was conducted off Terceira Island, Azores (Portugal) between May and August of 2012–2019. Shore- and vessel-based observations were conducted to locate and track groups of Risso's dolphins (*G. griseus*). Acoustic and movement data of seven adult individuals were collected using suction cup-attached DTAGs V.3 (240 kHz sound, 200 Hz accelerometer, magnetometer and depth data; [17]). The dolphins were tagged from a 6 m rigid-hulled inflatable vessel, using a 6–8 m carbon-fibre pole. Tags were placed between the blowhole and dorsal fin, dorsally or on the flank. Foraging dives were filmed using an unpiloted aerial system (DJI Phantom 4 Pro) to record the movement behaviour of animals at the surface [18]. The onset of a spin dive is an active surface behaviour (near-surface acceleration plus rotation, inducing a marked trail of white water) that can be reliably characterized from visual observations (figure 1; electronic supplementary material, video S1, [16]). Focal follows of dolphin groups were conducted throughout daylight hours and recorded the presence/absence of spin dives as a function of time of day (2 min sampling interval; *sensu* [19]). The presence of non-spin foraging dives could not be assessed from visual observation data alone.

## 2.2. Acoustic analysis—identify foraging effort

All acoustic and movement data from the DTAGs were analysed using Matlab 2014b (Mathworks, MA, USA), using scripts from the DTAG toolbox (available from: www.soundtags.org and www.

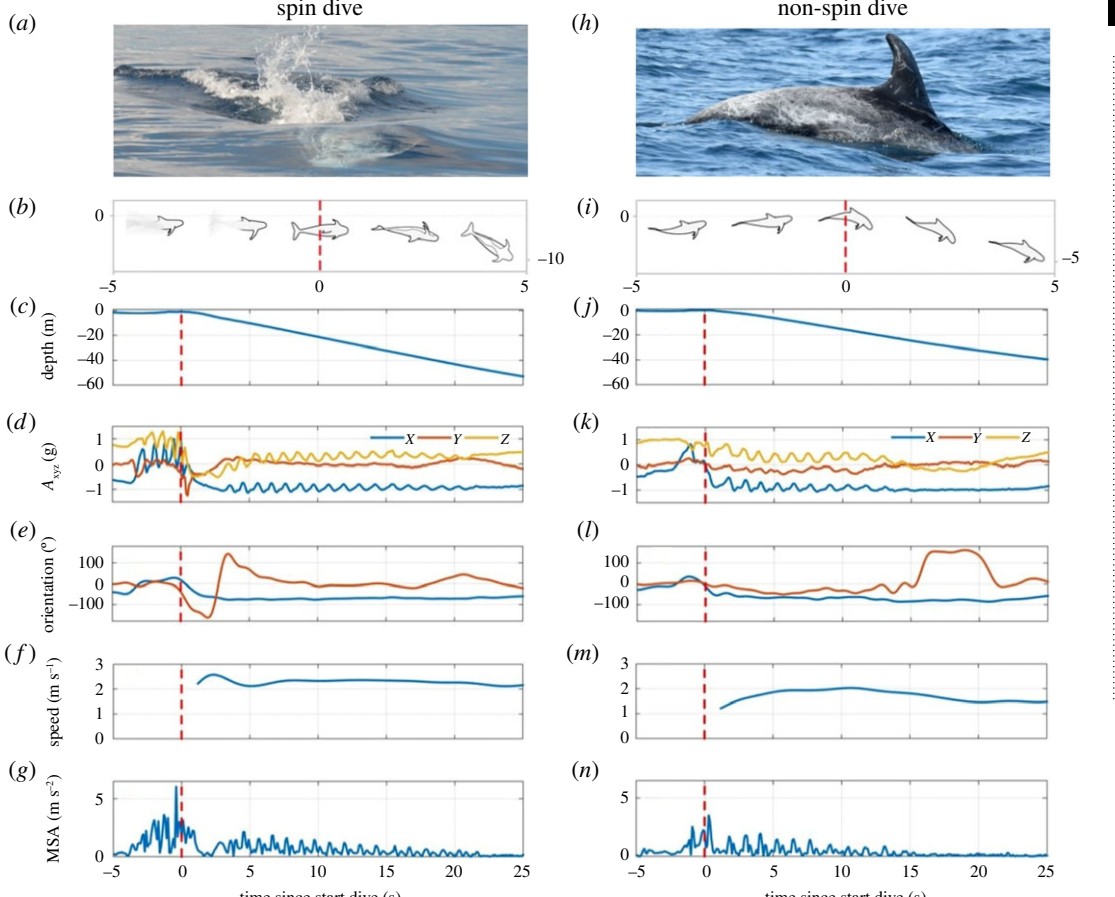

**Figure 1.** Fine-scale movement kinematics at the onset of spin and non-spin foraging dives. (*a–g*) Show a typical example of a spin dive and (*h–n*) of a non-spin dive (also known as 'arch-out dive'). Energetic spin (*a*) versus slow body arch (*h*) at the onset of a spin and a non-spin dive. (*b*) and (i): graphic representation of body orientation and depth from 5 s before to 5 s after the onset of the dive (indicated by the red vertical dashed line at $t = 0$). (*c–g*) and (*j–n*): movement kinematics from 5 s before to 25 s after the onset of the dive at $t = 0$. (*c,j*) Depth of the individual animal; (*d,k*) acceleration ($A_{xyz}$) along the three body axes (proxy for fluking), gliding initiated at 21 s (*d*) and 14 s (*k*); (*e,l*) change in body orientation, by pitch (blue) and lateral rotation (red); (*f,m*) forward speed; (*g,n*) MSA (proxy for energetic expenditure). In the spin dive, surface acceleration from strong fluking effort (onset at $t = -4$ s) is followed by a right-sided rotation (*e*; onset at $t = -1$ s), resulting in rapid descent (*c,f*). The individual performs a full spin (360° rotation, $t = -1$ to $t = +8$ s). Surface acceleration (*k,n*) and strong axial rotation (*l*) are absent in the first seconds of non-spin dives.

animaltags.org). The presence of foraging effort was identified from the biosonar vocalizations used by cetaceans to locate prey (echolocation clicks), recorded on the tag hydrophones. The timing of the start and end of echolocation click series (prey search vocalization) as well as foraging buzzes (vocalization at end of click series, indicating prey capture attempt; [20]) of the tagged animal were obtained manually through customized auditing scripts. Clicks by tagged dolphins can be readily distinguished from clicks produced by nearby conspecifics by their fairly consistent angle of arrival [20] and the existence of low-frequency energy (less than 15 kHz), which is absent in clicks produced by nearby animals [21,22]. The start and end of echolocation click series and the use of foraging buzzes were then matched to the depth of the tagged dolphins, using the pressure sensor data.

## 2.3. Identifying dive types

Visual observations indicated that Risso's dolphins off the Azores exhibited at least two dive types. In the first dive type, individuals speed up just below the surface after which they rotate onto their side before diving (spin dive; not previously described; figure 1*a*). In the second dive type, the animal arches its body out of the water, sometimes followed by the fluke breaching the surface (known as 'arch-out dive'; figure 1*h*). To identify these two dive types, we investigated pitch (descent/ascent angle) and lateral rotation (roll) during dives from the tag data. We computed the start times (*t*) of all dives greater than 20 m using the *finddives* function. Dives shallower than 20 m were excluded since these typically

represent regular breathing dives instead of foraging dives [11]. In addition, we excluded (i) the first dive of a deployment (possibly responding to tagging), (ii) incomplete dives that occurred at the end of a deployment during which the tag came off, (iii) dives during which the accelerometer signal was very noisy (i.e. not all suction cups well-attached), and (iv) dives that occurred in periods where the tag position could not be determined reliably.

The pitch and lateral rotation (degrees) of the tagged animals were computed from the raw accelerometer data on the tag following standard methods for tag data analysis [17]. Movement from accelerometer data can be separated into static and dynamic acceleration. Static movement describes low-frequency movement such as body orientation, while dynamic movement describes high-frequency movement like rapid stroking. We applied a low-pass filter to the pitch and rotation data to enable tracking of the body orientation at the start of each dive, using the *dsf* function. In order to remove all dynamic movement (related to stroking), the filter cut-off frequency was set at 70% of the dominant stroke frequency (e.g. [23]). The low-pass pitch and lateral rotation data were analysed for a 1 s window around the start of each dive (from $t = -0.5$ s to $t = +0.5$ s). For each window, we obtained the mean change in pitch (deviation from horizontal plane) and the value of rotation that deviated most from 0 (negative values indicate a right-sided rotation, positive values indicate a left-sided rotation). Dives were classified as spin dives if the mean change in pitch was lower than 0° and if the lateral rotation exceeded 60° in either direction (negative for right, positive for left). The threshold of 60° was chosen after inspection of the distribution of the maximum and minimum rotation values obtained for each window. Dives were classified as foraging dives if they contained one or more foraging buzzes. We then combined movement and acoustic data to classify four dive types: (i) spin dive with foraging (S + F+), (ii) spin dive without foraging (S + F−), (iii) non-spin dive with foraging (S − F+) and (iv) non-spin dive without foraging (S − F−). Subsequent analyses were conducted on the foraging dives (non-foraging dive characteristics reported in the electronic supplementary material, table S1).

## 2.4. Characterizing dive types

We analysed the duration, depth, movement metrics, echolocation characteristics and % foraging effort of each foraging dive. The start and end of foraging were defined as the timing of the first and last buzz in a dive, respectively, from which the time allocated to foraging was calculated. We defined the descent phase as the period from the start of a dive until the first buzz and the ascent phase as the period from the last buzz until the end of a dive, thus defining transit to and from prey layers, respectively. For the descent and ascent phase, we calculated the mean pitch (degrees) and the mean forward speed (m s$^{-1}$). Forward speed is defined as the speed in the direction of motion of the animal. It was estimated from the vertical speed (change in depth over time), divided by the sine of the pitch angle, using the orientation-corrected depth rate (*ocdr*) function [24]. This method becomes unreliable at low pitch angles and was therefore only calculated when the pitch angle of the animal exceeded 20°. We computed the mean minimum specific acceleration (MSA, in m s$^{-2}$; [25] for a 5 s window before (pre; $t = -5$ s) and after (post; $t = +5$ s) the onset of a dive. The MSA is a measure for the magnitude of dynamic (high-frequency) acceleration and can therefore be used as a proxy for stroking effort. The inter-dive-interval (IDI) was computed for all consecutive spin dives and non-spin dives that were spaced maximally 30 min apart ($N = 109/121$ total IDIs). To compare potential foraging gain over multiple dives between spin and non-spin dives, foraging bouts were identified as two or more consecutive foraging dives of the same type (spin or non-spin) with a maximum IDI of twice the respective mean dive duration (spin: 19.2; non-spin: 12.8 min). Differences between spin and non-spin foraging dive types and bouts were analysed using generalized linear mixed models (GLMMs; lme4 package; [26] with animal ID as a random effect, in R v. 3.4.3 [27]).

## 2.5. Prey records

Recording of potential prey in the water column was conducted using dual deployment of 38 and 120 kHz scientific echo sounders (Simrad EK60) from the research vessel [28].

Records were made in July/August 2018 and 2019 in Risso's dolphin foraging habitat, between 06.00 in the morning and 02.30 at night (1–6 h per record, during 10 days). Foraging habitat was previously defined from the visual observation of spin dives, during focal follows [16]. Upper and lower boundaries of the main biomass aggregation of apparent scattering layers were determined for each 30 min of recording time using Echoview (v. 9) and averaged to obtain mean depths for each 30 min period.

**Table 1.** Number of spin and non-spin foraging (F+) and non-foraging (F—) dives recorded during eight tag deployments (DTAGs) on seven adult Risso's dolphins (*Grampus griseus*), off Terceira Island, Azores.

| | | | | | spin | | non-spin | |
|---|---|---|---|---|---|---|---|---|
| ind. | tag ID | date of tagging | time of tagging | duration (h) | F+ | F— | F+ | F— |
| 1 | gg13_238a | 26 Aug 2013 | 16.12 | 5.7 | 2 | 0 | 3 | 8 |
| 1 | gg17_203a | 22 July 2017 | 12.33 | 9.4 | 20 | 0 | 0 | 5 |
| 2 | gg15_229a | 17 Aug 2015 | 9.41 | 16.7 | 10 | 3 | 30 | 22 |
| 3 | gg15_229c | 17 Aug 2015 | 13.53 | 11.0 | 0 | 0 | 18 | 19 |
| 4 | gg16_169a | 17 June 2016 | 14.12 | 4.9 | 8 | 2 | 3 | 8 |
| 5 | gg16_171a | 19 June 2016 | 8.51 | 11.8 | 4 | 1 | 0 | 12 |
| 6 | gg17_200a | 19 July 2017 | 11.04 | 15.9 | 25 | 1 | 1 | 10 |
| 7 | gg18_214a | 2 Aug 2018 | 15.37 | 10.2 | 6 | 0 | 15 | 30 |
| Total | | | | 85.6 | 75 | 7 | 70 | 114 |

# 3. Results and discussion

## 3.1. Risso's dolphins employ two foraging dive strategies

We analysed the acoustic and movement behaviour of eight tag recordings from seven individual Risso's dolphins, performing a total of 266 dives, reaching depths of 20–623 m (table 1). Spin dives produced a distinct signal in the accelerometer data and could readily be distinguished from non-spin dives (figure 1; electronic supplementary material, figure S1). Spin dives were initiated by strong fluking (figure 1*a,d*) followed by a pronounced right-sided lateral rotation (63–171° in the 1 s window centred around the start of each dive; figure 1*a,b,e*; electronic supplementary material, figure S1 and video S1). After the initial spin, the animals rapidly descended into deeper water (figure 1*c,f*) at a steep, downward-angled pitch (figure 1*e*; electronic supplementary material, figure S1). The lateral rotation after the initial spin at the surface varied across dives, individuals either kept rotating, up to or over 360°, maintained their lateral orientation or gradually rotated back left. In contrast, non-spin dives lacked strong fluking at dive-onset and started with a change in pitch (figure 1*h,i,k,l*), indicative of an arching of the body as typically observed in cetacean dives.

In total, 82 of the 266 dives were spin dives, performed by six out of seven individuals. Of these, 75 spin dives (91%) contained foraging buzzes (table 1). Non-foraging dives were predominantly non-spin dives (94%). These results confirm the hypothesis that Risso's dolphin spin dives are foraging dives. In total, seven spin dives (9%) did not contain foraging buzzes, possibly representing aborted foraging dives. Three of these dives were the only left-sided spin dives. Spin foraging dives were always right-sided, an interesting example of lateralization in marine mammals (electronic supplementary material, figure S1). However, Risso's dolphins did not forage exclusively during spin dives: 38% of non-spin dives contained foraging buzzes (table 1).

## 3.2. Spin dives: getting deep, fast

Spin dives were more than two times deeper than non-spin foraging dives. More specifically, on average the maximum depth per spin dive was 426 m whereas that of non-spin dives was only 178 m (table 2). Similarly, the mean depth of prey capture attempts (buzzes) was 344 m for spin dives but only 148 m for non-spin dives (table 2). Furthermore, spin dives had a significantly higher forward speed during descent than non-spin dives (2.5 versus 2.0 m s⁻¹) and a steeper downward angle (−60° versus −41°; table 2 and figure 1). As a result, while first prey capture attempts were made significantly deeper during spin dives than non-spin dives (287 versus 126 m), there was no significant difference in time spent to reach first prey (145 versus 121 s; table 2). Spin dives thus represent a dive strategy to obtain fast access to deep-dwelling prey.

Risso's dolphins actively stroked during the first part of their spin foraging dive descent, performing a lateral rotation at dive-onset (the spin). After reaching negative buoyancy, this was followed by

**Table 2.** Characteristics of adult Risso's dolphin (*Grampus griseus*) spin and non-spin foraging dives. Values represent mean (±s.d.) per dive, and GLMM test results for significance of difference between spin and non-spin dives and bouts. MSA: minimum specific acceleration (m s$^{-2}$; mean for 5 s pre- and 5 s post-dive-onset). IDI: inter-dive-interval.

| | foraging (F+) | | GLMM results | | |
|---|---|---|---|---|---|
| | non-spin (S−) | spin (S+) | estimate (s.e.) | t | p |
| dive time (min) | 6.4 (1.6) | 9.6 (2.1) | 3.0 (0.4) | 7.8 | <0.0001 |
| maximum depth (m) | 177.8 (76) | 425.7 (132) | 265.0 (20.9) | 12.7 | <0.0001 |
| descent speed (m s$^{-1}$) | 1.98 (0.4) | 2.47 (0.5) | 0.7 (0.1) | 7.0 | <0.0001 |
| ascent speed (m s$^{-1}$) | 2.18 (0.4) | 2.30 (0.4) | 0.3 (0.1) | 3.9 | 0.0001 |
| descent angle (degrees) | −41.4 (16.5) | −59.9 (13.5) | −18.4 (3.1) | −6.0 | <0.0001 |
| ascent angle (degrees) | 46.5 (16.1) | 68.6 (11.0) | 23.1 (2.8) | 8.3 | <0.0001 |
| mean pre-MSA (m s$^{-2}$) | 0.64 (0.17) | 1.99 (0.97) | 1.0 (0.1) | 8.2 | <0.0001 |
| mean post-MSA (m s$^{-2}$) | 0.80 (0.21) | 1.00 (0.55). | 0.1 (0.1) | 1.9 | 0.06 |
| IDI (min) | 10.0 (4.8) | 14.7 (3.6) | 4.8 (0.9) | 5.6 | 0.0002 |
| no. buzz | 5.5 (3.5) | 10.7 (7.9) | 5.4 (1.2) | 4.4 | 0.0001 |
| depth start clicking (m) | 15.5 (9.7) | 90.2 (76.5) | 92.1 (10.8) | 8.6 | <0.0001 |
| time to start clicking (s) | 7.7 (3.9) | 35.6 (26.6) | 31.3 (3.7) | 8.3 | <0.0001 |
| foraging time in dive (%) | 42 (23) | 49 (16) | 7.3 (3.2) | 2.3 | 0.03 |
| buzz depth (m) | 148.1 (56.7) | 344.2 (122.9) | 196.8 (18.1) | 10.8 | <0.0001 |
| depth first buzz (m) | 126.1 (55.5) | 287.0 (129.6) | 154.3 (20.5) | 7.5 | <0.0001 |
| time to first buzz (s) | 120.5 (71.6) | 144.8 (60.9) | 23.5 (11.5) | 2.1 | 0.053 |
| inter-buzz-interval (s) | 45.8 (45.5) | 49.8 (40.9) | 4.0 (7.3) | 0.5 | 0.59 |
| buzz rate bout (min$^{-1}$) | 0.7 (0.2) | 0.9 (0.6) | 0.3 (0.2) | 1.4 | 0.17 |

effortless gliding (sinking, no fluking effort; figure 1), often including additional, slow rotation (mean (s.d.) 1.2 (1.2) rotations during glide descent). Gliding is a common strategy among deep divers, serving to conserve cost of transport [9,29,30]. The animals resumed stroking at the onset of active foraging (first buzz; electronic supplementary material, figure S2). Rapid descent was matched with enhanced ascent speed and angle, as compared to non-spin dives, enabling a larger portion of the dive to be spent foraging (49 versus 42% in spin and non-spin dives; table 2). Gliding, with possible slow rotation (mean (s.d.) 0.7 (0.9) rotations), was also performed during non-spin dive descents, following initial stroking and up to the first prey capture attempt (figure 1; electronic supplementary material, figure S2). The marked exhalation at onset of the spin dive (electronic supplementary material, video S1) suggests that individuals also release air volume to further reduce buoyancy. Combined, these data indicate that during spin dives, individuals aim to initiate their glide with as high as possible speed, in order to descend as quickly as possible.

Spin dives were only employed to reach deep prey. Moreover, during spin dives, Risso's dolphins did not probe surface waters for prey and only started biosonar prey search (clicking) at an average depth of 90 m at 36 s into the descent. This is significantly deeper (90 versus 16 m) and significantly later (36 versus 8 s; table 2) than the near-surface start of clicking during non-spin foraging dives. These observations confirm the finding that Risso's dolphins plan their target foraging zone, prior to dive-onset (figure 1) [14].

## 3.3. Spin dive energetics

Whereas spin dives were significantly longer than non-spin dives (9.6 versus 6.4 min; table 2), both dive type durations remained well within the species' calculated aerobic dive limit (cADL; 14.8–16.2 min for adults [11]), with a comparable relationship between dive duration and IDI across dive types (mode: IDI = 1.3× dive duration). Thus, Risso's dolphins mainly foraged aerobically, a strategy maximizing animal long-term foraging efficiency [29]. As identified by the MSA, a proxy for energy expenditure, the spin is a brief but energetically costly rotation-sprint. MSA just prior to dive-onset was 2.0 m s$^{-2}$ for spin dives but only 0.6 m s$^{-2}$ for non-spin dives (figure 1 and table 2). The rate of oxygen consumption is predicted to

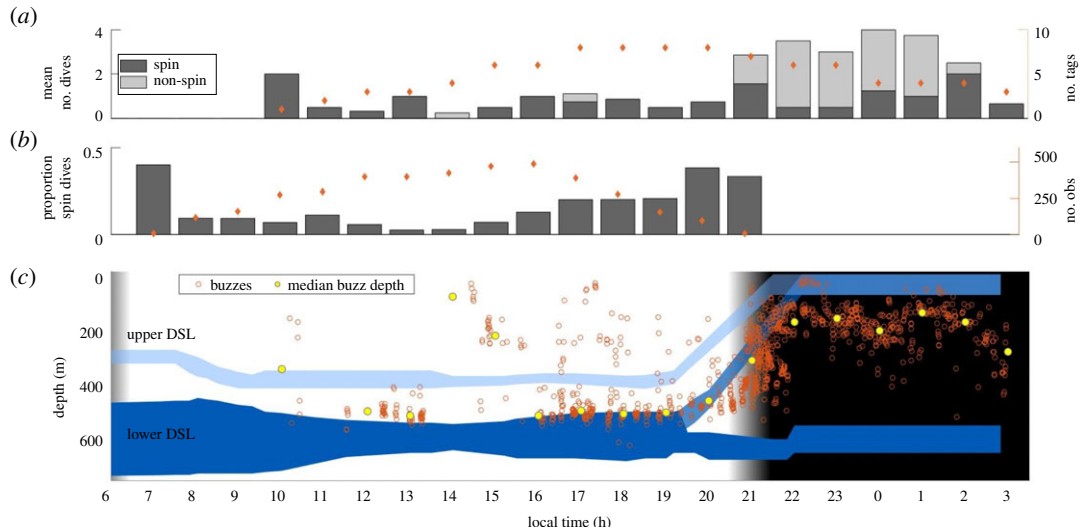

**Figure 2.** Risso's dolphin foraging dives track the diel vertical migration of the deep scattering layer (DSL). Diel pattern of (*a*) the number of spin and non-spin foraging dives of the tagged individuals (tag data; orange circles: number of tagged individuals), (*b*) the proportion of observation records with spin dives (visual observation data; orange circles: number of observations (obs)), (*c*) depth of prey capture attempts, as identified by buzzes of the tagged individuals, tracks the depth of the DSL (blue layers, showing the depth contour of the upper and lower DSL). (*a*) Timing of tag records (10.00–03.00) is determined by the period of tag attachments; (*b*) period of visual observation records (7.00–21.00) is confined to daylight hours. Note: (*b*) $n = 5$ and $n = 6$ visual observations at 7.00 and 21.00.

increase cubically with swim speed [4]. Consequently, energetically costly behaviours such as high motility and sprint are rarely encountered during transit. Their use would require augmented, reliable energetic rewards from prey ('spend more—gain more' [31]). To minimize cost of transport, dive strategies in most cetacean species are characterized by low speeds during transit, in combination with periods of gliding [9,30–33]. The spin dive deviates from this strategy, by adding a short but dynamic surface spin to increase overall descent speed (figure 1; electronic supplementary material, figure S2), allowing for quick arrival and more time to be spent at depth.

## 3.4. Foraging Risso's dolphins trace the deep scattering layer

Dolphins preferred shallow nocturnal foraging, starting around dusk. In total, 76% of foraging dives recorded on the tags were performed after 21.00. Less common occurrence of daytime foraging (10% of foraging dives recorded on the tags) was confirmed by the visual observation records (figure 2*a,b*). Spin dives were observed from morning to late afternoon, with clearly higher rates of occurrence around dusk and dawn (figure 2*b*). Foraging Risso's dolphins actively targeted, and traced, the deep scattering layer (DSL; figure 2*c*). DSLs occur throughout the world's oceans and represent high-density aggregations of multiple taxonomic groups, including cetacean prey (e.g. fish, cephalopods and crustaceans) [34,35]. Two main DSLs were observed in the area, showing typical patterns of diel vertical migration [34,36]: a lower DSL situated approximately 500–700 m deep during daytime and an upper, narrower DSL situated at approximately 400 m depth. Animals from both layers migrated to surface waters around dawn (figure 2*c*). Risso's dolphin foraging dive depth closely traced the diel variation in DSL depth, facilitated through adjustment of their foraging strategy. Individuals showed a marked shift from predominant daytime use of deep spin dives (9.00–21.00; spin dives: 92% of total foraging dives) to the use of shallow, non-spin dives around dusk (later than 21.00; non-spin dives: 63% of total foraging dives) (figure 2*a*). Only 6% of non-spin foraging dives were performed during daylight hours.

Switching between dive types appears to be an established strategy employed by Risso's dolphins to reach prey layers at different depth zones (figure 2, [15,16]). Risso's dolphins off Santa Catalina Island, California, foraging in an entirely different ecological setting, showed regular switches between shallow and deep foraging throughout the day, across four static or migratory scattering layers, centred around 50–425 m. In spite of the differences in distribution of foraging effort over the day, and in target prey fields between the two areas, individuals observed off California also significantly increased descent angle and forward speed, and doubled their prey capture attempts when switching from shallow to deep

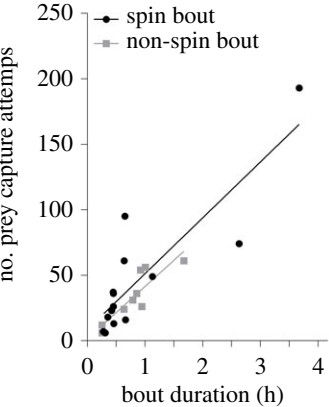

**Figure 3.** Potential foraging gain of Risso's dolphin spin and non-spin foraging bouts. The number of prey capture attempts (buzzes) per hour of foraging effort identify comparable foraging rates between spin and non-spin foraging bouts (linear regression; spin bout: $R^2 = 0.74$, $p = 0.002$; non-spin bout: $R^2 = 0.76$, $p < 0.0001$).

dives [11,15]. Whereas the presence of spins was not mentioned, visual observations confirmed that individuals off California perform similar spin dives as at the Azores (F.V. July–September 2011–2014, personal observation).

## 3.5. Benefits of deep dives?

Fish predators of the DSL commonly trace its diel movement [37]. The few delphinid predators that have been reported to prey on the DSL, however, predominantly target the shallow night-time period [35,38]. To also enable exploitation of the deep-dwelling DSL, Risso's dolphins have evolved the spin dive strategy. Spin dives effectively minimize the transit time from the surface to deep prey layers, facilitating deep mesopelagic foraging. Deep and long dives, however, are energetically costly [4], which is further augmented by applying sprints. As central place foragers, Risso's dolphins can only accommodate longer, more costly travel if this is balanced with enhanced energetic return from their prey [39,40]. The benefit of deep foraging zones may lie in the near-doubling of prey capture attempts in spin versus non-spin foraging dives (number of buzzes: 10.7 versus 5.5; table 2), suggesting the presence of bountiful prey at depth. Ample foraging opportunity on deep-dwelling prey could balance a high-cost travel strategy. Foraging, however, typically occurred in bouts of multiple consecutive foraging dives ($N = 24$ bouts, comprising 2–15 dives, duration: 9–221 min). IDIs were significantly longer between spin dives than between non-spin dives (14.7 versus 10.0 min, table 2). This suggests that spin dives require a longer recuperation period than non-spin dives. The rate of prey capture attempts did not differ significantly between foraging bouts comprising spin dives and bouts of non-spin dives (0.9 versus 0.7 buzzes per bout-minute; table 2 and figure 3). Hence, over multiple dives, the significantly enhanced foraging potential of a spin dive was counterbalanced by the longer duration and IDI of spin versus non-spin dives (table 2). The rate of prey capture attempts did not differ significantly between spin and non-spin dives, suggesting comparable target prey fields (inter-buzz-interval: 50 versus 46 s; table 2). Conversely, the targeting of significantly larger or more calorific prey during deeper dives would probably result in a marked increase in prey search and handling times (increased inter-buzz-interval) and reduced overall capture rates, as demonstrated by deep versus shallow foraging on few versus many prey in pilot whales (*Globicephala macrorhynchus*) [41]. If Risso's dolphins indeed forage on similar prey while tracing the diel vertical migration of the DSL, the rate of prey encounter during the foraging bout may be the 'currency' that is optimized, driving foraging decisions such as foraging onset and dive strategy choice [3].

The leading hypothesis for the existence of the high-biomass aggregations of the DSL is that it offers protection from visual predators [37,42]. Cetaceans, however, commonly forage using sound (echolocation) [8] and do not depend on light to capture prey. Still, whereas Risso's dolphins can exploit the DSL throughout the day, individuals showed a preference for deep foraging around dusk and dawn (figure 2a,b). This timing is associated with the respective upward and downward migration of the DSL. It indicates that for deep dives, Risso's dolphins may specifically target periods during which their ectothermic cephalopod prey shows pronounced shifts in behaviour, school density and physiological capabilities. Environmental conditions (e.g. temperature, light) and cephalopod schooling behaviour vary as

a function of depth and time of day. This modulates their catchability through variation in escape responses, density and vigilance [43–45]. In line with these expectations, both spin and non-spin dive buzz rates were significantly higher during dusk, than during daytime (18.6 versus 10.3 buzzes per minute for spin dives; GLMM, estimate (s.e.): 4.1 (2.1), $t = 2.0$, $p = 0.05$) and night-time (8.6 versus 4.9 buzzes per minute for non-spin dives; estimate (s.e.) = 1.8 (0.9), $t = 2.1$, $p = 0.04$), respectively. Around dusk, cephalopod prey transitions from predator-avoidance behaviour to migration, while still residing in relatively high densities and in colder waters that constrain its capacity for swift escape. Conversely, around dawn, prey enters colder waters prior to reaching its refuge [44]. The daily vertical migrations may therefore represent pivotal periods during which the higher costs of transport to remote layers are counterbalanced by ease of prey capture.

# 4. Conclusion

Whereas Risso's dolphins off the Azores were mostly nocturnal foragers benefitting from near-surface prey, they have evolved an energetic spin dive strategy to effectively target deep-dwelling prey. Energetically costly manoeuvres and sprints are typically employed by predators during the final stages of a hunt, when energetic reward of prey capture is imminent [25,41,46]. By contrast, Risso's dolphins used high-cost manoeuvres at the onset of travel to remote prey. This signifies that individuals performing spin dives can rely on sufficient energetic return, possibly from being able to target a dependable, dense and high-biomass prey layer such as the DSL. Our findings shed new light on how air-breathing marine predators can maintain optimal foraging performance when targeting remote prey and actively plan access to deep-sea resources. Risso's dolphin proficient exploitation of both near-surface and deep-sea prey signifies the role of cetaceans as key drivers of oceanic food web dynamics, and reveals a direct ecological linkage between deep and shallow systems.

Data accessibility. Data supporting this manuscript are accessible in the Dryad Digital Repository at: https://doi.org/10.5061/dryad.c2fqz6173. The data are provided in the electronic supplementary material [47].
Authors' contributions. F.V., O.A.K and M.G.O. conceptualized the study and designed methodology; F.V., M.G.O., O.A.K. and A.C.M.K. conducted the investigation process; O.A.K. and F.V. performed formal analysis; O.A.K., F.V. and E.H.M.S. drafted the manuscript; F.V., J.H. and E.H.M.S. provided scientific supervision and acquired funding; D.P.N. provided essential resources; all authors critically revised the manuscript. All authors gave final approval for publication and agree to be held accountable for the work performed therein.
Competing interests. The authors declare no competing interests.
Funding. This work was supported by the Office of Naval Research, USA (ONR; award nos. N00014-15-1-2341 and N00014-17-1-2715) and by the Dutch Research Council (NWO; Veni grant no. 016.Veni.181.086).
Acknowledgements. We gratefully acknowledge the support of all field team members, in particular Charlotte Curé, Ricardo Antunes, Luis Barcelos, Stacy DeRuiter, Francisco Reis, Ricardo Fernandes and the OceanEmotion team, and Sanne Hessing for aid in acoustic analysis. We thank Prof. Peter Tyack for the use of tagging equipment, Brandon Southall (SEA) for collaboration with the SOCAL-BRS project, Prof. Eduardo Brito de Azevedo and Francisco Reis (ITTAA, University of the Azores) for use of the research vessel Atlantida, and Profs. Rosalina Gabriel, Paolo Borges and Joao Pedro Barreiros of GBA (CE3C), University of the Azores for their collaboration.

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
