## [Peer Review File · Royal Society Open Science]

Review History

RSOS-202320.R0 (Original submission)

Review form: Reviewer 1

Is the manuscript scientifically sound in its present form?

Yes

Are the interpretations and conclusions justified by the results?

Yes

Is the language acceptable?

Yes

Do you have any ethical concerns with this paper?

No

Have you any concerns about statistical analyses in this paper?

No

Recommendation?

Accept with minor revision (please list in comments)

Comments to the Author(s)

This is a nice and well-written story that I greatly enjoyed reading. However, I do believe that this story can be completed by some extra analyses and a bit of theoretical background regarding the optimal diving theory to be made stronger. You will find below my main comments and suggestions.

L207: not sure about what you mean by greater forward speed (there is the absolute swimming speed and the vertical speed).

So is the greater vertical speed in spinning dives greater is due to a steeper pitch (descent angle) and a greater swimming speed or just a greater descent pitch? So is the initial acceleration aiming at reaching a greater absolute swimming speed when the Risso dolphins are gliding down on the spinning dives?

L210: You never mention in your paper the Aerobic Dive Limit (or the Behavioral ADL. Could you estimate what is the difference in diving expenditure and on ADL between the two types of deep dives. Are Risso exceeding their bADL when performing spinning dive. Just plot the surface duration consecutive to the dive duration, you could also include an index of overall swimming effort through the dive.

Does the relationship between dive duration and the post dive interval varies (or not) between spinning and non-spinning dives? This should be nice to look at.

L239 : This is the information I was missing§ I was wondering if the Risso Dolphins were actively stroking through the whole dive or not . So is the idea is to increase the initial speed as much as possible prior to the gliding phase to descend as quickly as possible, but this swimming effort should last until the Risso are fully compressed and become negatively buoyant (to glide down). May be that what is the biggest constrain on these animal is the amount of oxygen stores and therefore this is the best strategy allowing them to spend more time at depth for a small increase of energy expenditure at the beginning of the dive, I guess this could be modelled. See also the work performed on pilot whales in the Strait of Gibraltar.

I would encourage you to describe more precisely the different phases of the dive, we can see for you figure 1 that the Risso stroke for approximately 20 seconds at the beginning of the dive and then they glide. However, this does not tell us the depth at which they are nearly fully compressed and became negatively buoyant to glide down.

So I wonder if the diving strategy in spinning dives is to initiate the glide with a greater initial speed to go down as quickly as possible, you have all the elements to verify that and to state it clearly if it is the case. In my view the dive strategy could be described more precisely and the whole story will be easier to follow, mostly for people who are not familiar with diving behavior studies.

L239: I do believe that you have all the information necessary to investigate if Risso are targeting different prey items between day and night and during the day between spinning dives and non-spinning ones. Have you been looking at the echoes of the clicks that you should be able to get on the D-tag, do they tell you something about possible differences in targeted prey's sizes between the two dives categories with the hypothesis that when performing spinning dives Risso might be targeting more rewarding prey (i.e. larger preys) compared to non-spinning dives. Just a guess. Does the chase behavior in terms of swimming effort, direction changes of the prey is identical or differ between those two situations. If I understood well what you suggest, during the day preys are deeper and in colder waters and as they are ectotherms, they are likely to be

less active compare to the night when preys are closer to the surface in warmer water (what is the temperature gradients).

L255 change in the pitch of the descent in relation to the foraging success of the previous dive has been found in many diving seabirds and marine mammals equipped with accelerometers and pressure sensor. Most of the time they increase their dive angle to return more quickly at depth and maximize the bottom duration of their dive (the main foraging phase of the dive). You neither refer to the bottom duration of the dive and I would encourage you to do so to verify how this change according to diving depth and between spinning versus non spinning dives. L277 ok but this should be expressed per unit of time spent diving + recovering (not per dive) as indicated above differences in targeted prey sizes might be part of the equation and should be discussed.

Review form: Reviewer 2

Is the manuscript scientifically sound in its present form?

Yes

Are the interpretations and conclusions justified by the results?

Yes

Is the language acceptable?

Yes

Do you have any ethical concerns with this paper?

No

Have you any concerns about statistical analyses in this paper?

No

Recommendation?

Accept with minor revision (please list in comments)

Comments to the Author(s)

See attached (Appendix A).

Review form: Reviewer 3

Is the manuscript scientifically sound in its present form?

Yes

Are the interpretations and conclusions justified by the results?

Yes

Is the language acceptable?

Yes

Do you have any ethical concerns with this paper?

No

Have you any concerns about statistical analyses in this paper?

No

Recommendation?

Major revision is needed (please make suggestions in comments)

Comments to the Author(s)

I feel this manuscript will contribute greatly to advancing scientific literature and increasing our knowledge regarding cetacean diving behavior and its role in marine ecosystems. To summarize the manuscript, the authors aimed to characterize a unique dive type (termed a spinning dive) used by Risso's dolphins of the Azores and interpret the function of this dive strategy. The authors hypothesized this newly described spinning dive is utilized to target deep-dwelling prey and to optimize foraging performance. The authors tested this hypothesis using tag data from seven Risso's dolphins and compared the behavior and kinematics between spinning and non-spinning dives. The main take-away from this study is that Risso's dolphins both proactively plan and then utilize a metabolically costly spinning dive to reach deep-dwelling prey, and while it poses an energetic risk, this dive strategy is sufficiently rewarded with access to a densely populated prey environment.

I believe the authors were thorough when justifying reasoning for their methods and that the statistics are sound. While reading, I found the writing was concise and clear (except for a minor few instances, see suggestions line by line). The structure is well-organized, and the manuscript is a good length. I believe the topic is remarkably interesting and the authors connected their findings into a gap in knowledge of cetacean diving strategies. While the manuscript is satisfactory, I feel there are changes that should occur prior to publication.

I would like to address my suggestions of this manuscript, in order of each section, line-by-line.

Background

62 and 63 – Revise for flow, maybe consider combining the first two sentences.

70 – When I hear the word spinning, I think of a spinner dolphin or spinner shark, both of which make several 360 revolutions while spinning. From Figure 1 and the supplementary video, it appears the dolphins may only complete one spin. Additionally, it is not mentioned in the text if they spin multiple times. I found in the supplementary figure 2 legend that the represented individual performed two spins. I strongly suggest the following edits to resolve this concern.

1) Mention in the text the finding of multiple rotations while diving (I would suggest providing as an average across all individuals) because without the supplementary information directly in front of the reader, the manuscript suggests otherwise. The term “spinning” dive is misleading if the dolphins only complete one revolution. By mentioning in the text (and suggestions for figures mentioned later) it will be apparent to the reader they spin multiple time.

2) I am curious if the individual that performed two rotations is an average number of spins. Figure S2 is representative of one individual so if the average number of spins across all animals is around one, to term the dive “spinning” is misleading. If you provide the number of spins and clarify in the text it will greatly support your naming of this new dive strategy.

3) To further clarify this concern, if you have video that shows a dolphin diving and completing a full rotation, (I am assuming unlikely since they are too deep to see from a drone at this point in the spin) it would be more representative of a “spinning” dive. From the video, it looks more like a roll.

Methods

84 – You state the data comes from 7 individuals; I would suggest clarifying if you used one dive sequence (descent and subsequent ascent) per animal or otherwise. Later in the results it states 8 recordings from 7 individuals, I suggest clarifying if one individual was tagged twice or if you used two separate recordings from the same tag.

87 – “Tags were generally...” – if there was an individual in the study that had its tag in a different location, I suggest it be addressed in the methods. If not, I do not think it’s necessary to say “generally”.

117 – Consider rewording for sentence flow, the end of the sentence is unclear.

119 – I am confused if the low-pass and high-pass filter was applied in on an additional software or in Matlab. If other software was used, you should expand the methods to include it. Otherwise, consider rewording for clarification.

123 – I think the methods section would be strengthened if you clarify why the filter-cut off frequency was set at 70%

Results

217 – Consider combining “This spin is...” with “This is shown” to better flow.

Figures

Figure 1 – For consistency reasons and to help with the comparison between dive types, I believe it would be easier to interpret if parts a,b,h,i were to be made into a separate figure. I suggest keeping the graphical representation (b and i) but using multiple stills to replace parts a and h. The single photograph used in part a does not contribute a lot of information about the spin behavior, however if you used several stills in a sequence above the drawing (b), it would be concise and clear to the reader. Doing the same for the non-spinning dive next to it would allow for easier comparison.

I believe the latter part of figure 1 (parts c-g and j-n) as presented can then be replaced by Figure S2 (rename it figure 2) because it shows a longer time scale and still represents your trends from figure 1 very well. Additionally, by making these graphs their own figure, you can also make them larger and easier to interpret.

Figure 2 – I think this figure is well made and should be made figure 3.

Figure 3 – I think this can be moved to supplementary material considering it is referenced once in the results and discussion section. Also, it can easily be described verbally by the text unlike your other figures that I believe are more visually representative of the results.

Table 1 – One individual did not perform any spinning dives and this respective animal was tagged just before 2pm local time and the tag was deployed for 11 hours. This is my pure curiosity and interest, but it would be satisfying as a reader if you presented any life history information (if you have any) in Table 1.

Video S1 – I think it would be beneficial to make the video a similar, side by side comparison like parts a and b of figure one. I suggest playing video side by side of a spin vs non spinning dive and insert the graphical representation in the corner/on the side to help readers visualize the movements. I personally do not think the half speed and associated text are necessary.

Overall, I recommend acceptance of this manuscript with major edits. I hope the authors will take my suggestions to further strengthen their publication. I am extremely excited about this manuscript; I think it will be greatly beneficial to the field of marine mammal diving behavior and ecology as it fills a gap in knowledge of marine predator foraging in the pelagic ocean.

Decision letter (RSOS-202320.R0)

Dear Dr Visser

The Editors assigned to your paper RSOS-202320 "Risso's dolphins perform spinning dives to target deep-dwelling prey" have now received comments from reviewers and would like you to revise the paper in accordance with the reviewer comments and any comments from the Editors. Please note this decision does not guarantee eventual acceptance.

Please submit your revised manuscript and required files (see below) no later than 21 days from today's (ie 14-Jun-2021) date. Note: the ScholarOne system will 'lock' if submission of the revision is attempted 21 or more days after the deadline. If you do not think you will be able to meet this deadline please contact the editorial office immediately.

on behalf of Professor Pete Smith (Subject Editor)
openscience@royalsociety.org

Associate Editor Comments to Author:

Please accept our apologies for the unusual delay in completing the review of your paper - as we're sure you can imagine, it has proved exceptionally hard in recent months to secure the support of reviewers for many of the papers handled by the editors. We are, therefore, extremely grateful to the three commentators who have provided such extensive feedback on your paper. While it seems your work is broadly on track for acceptance, there are a number of modifications recommended by the reviewers that we'd like you to make - hopefully they won't be too onerous to enact, but we'd like to give you sufficient time to make the changes (hence the 3-week deadline). As one of the reviewers comments that their recommendations are major, we will ask for their advice after receipt of your revision, but we hope this will be a quick turnaround. Thanks again for your support.

Reviewer comments to Author:

Reviewer: 1

Comments to the Author(s)

This is a nice and well-written story that I greatly enjoyed reading. However, I do believe that this story can be completed by some extra analyses and a bit of theoretical background regarding the optimal diving theory to be made stronger. You will find below my main comments and suggestions.

L207: not sure about what you mean by greater forward speed (there is the absolute swimming speed and the vertical speed).

So is the greater vertical speed in spinning dives greater is due to a steeper pitch (descent angle) and a greater swimming speed or just a greater descent pitch? So is the initial acceleration aiming at reaching a greater absolute swimming speed when the Risso dolphins are gliding down on the spinning dives?

L210: You never mention in your paper the Aerobic Dive Limit (or the Behavioral ADL. Could you estimate what is the difference in diving expenditure and on ADL between the two types of deep dives. Are Risso exceeding their bADL when performing spinning dive. Just like the surface duration consecutive to the dive duration, you could also include an index of overall swimming effort through the dive.

Does the relationship between dive duration and the post dive interval varies (or not) between spinning and non-spinning dives? This should be nice to look at.

L239 : This is the information I was missing. I was wondering if the Risso Dolphins were actively stroking through the whole dive or not. So is the idea is to increase the initial speed as much as possible prior to the gliding phase to descend as quickly as possible, but this swimming effort should last until the Risso are fully compressed and become negatively buoyant (to glide down). Maybe that what is the biggest constraint on these animals is the amount of oxygen stores and therefore this is the best strategy allowing them to spend more time at depth for a small increase of energy expenditure at the beginning of the dive, I guess this could be modelled. See also the work performed on pilot whales in the Strait of Gibraltar.

I would encourage you to describe more precisely the different phases of the dive, we can see from your figure 1 that the Risso stroke for approximately 20 seconds at the beginning of the dive and then they glide. However, this does not tell us the depth at which they are nearly fully compressed and became negatively buoyant to glide down.

So I wonder if the diving strategy in spinning dives is to initiate the glide with a greater initial speed to go down as quickly as possible, you have all the elements to verify that and to state it

clearly if it is the case. In my view the dive strategy could be described more precisely and the whole story will be easier to follow, mostly for people who are not familiar diving behavior studies.

L239: I do believe that you have all the information necessary to investigate if Risso's are targeting different prey items between day and night and during the day between spinning dives and non-spinning ones. Have you been looking at the echoes of the clicks that you should be able to get on the D-tag, do they told you something about possible differences in targeted prey's sizes between the two dives categories with the hypothesis that when performing spinning dives Risso's might be targeting more rewarding prey (i.e. larger preys) compare to non-spinning dives. Just a guess. Does the chase behavior in tem of swimming effort, direction changes of the prey is identical or differ between those two situations. If I understood well what you suggest, during the day preys are deeper and in colder waters and as they are ectotherms, they are likely to be less active compare to the night when preys are closer to the surface in warmer water (what is the temperature gradients).

L255 change in the pitch of the descent in relation to the foraging success of the previous dive has been found in many diving seabirds and marine mammals equipped with accelerometers and pressure sensor. Most of the time they increase their dive angle to return more quickly at depth and maximize the bottom duration of their dive (the main foraging phase of the dive). You neither refer to the bottom duration of the dive and I would encourage you to do so to verify how this change according to diving depth and between spinning versus non spinning dives. L277 ok but this should be expressed per unit of time spent diving + recovering (not per dive) as indicated above differences in targeted prey sizes might be part of the equation and should be discussed.

Reviewer: 2

Comments to the Author(s)

See attached

Reviewer: 3

Comments to the Author(s)

I feel this manuscript will contribute greatly to advancing scientific literature and increasing our knowledge regarding cetacean diving behavior and its role in marine ecosystems. To summarize the manuscript, the authors aimed to characterize a unique dive type (termed a spinning dive) used by Risso's dolphins of the Azores and interpret the function of this dive strategy. The authors hypothesized this newly described spinning dive is utilized to target deep-dwelling prey and to optimize foraging performance. The authors tested this hypothesis using tag data from seven Risso's dolphins and compared the behavior and kinematics between spinning and non-spinning dives. The main take-away from this study is that Risso's dolphins both proactively plan and then utilize a metabolically costly spinning dive to reach deep-dwelling prey, and while it poses an energetic risk, this dive strategy is sufficiently rewarded with access to a densely populated prey environment.

I believe the authors were thorough when justifying reasoning for their methods and that the statistics are sound. While reading, I found the writing was concise and clear (except for a minor few instances, see suggestions line by line). The structure is well-organized, and the manuscript is a good length. I believe the topic is remarkably interesting and the authors connected their findings into a gap in knowledge of cetacean diving strategies. While the manuscript is satisfactory, I feel there are changes that should occur prior to publication.

I would like to address my suggestions of this manuscript, in order of each section, line-by-line.

Background

62 and 63 – Revise for flow, maybe consider combining the first two sentences.

70 – When I hear the word spinning, I think of a spinner dolphin or spinner shark, both of which make several 360 revolutions while spinning. From Figure 1 and the supplementary video, it appears the dolphins may only complete one spin. Additionally, it is not mentioned in the text if they spin multiple times. I found in the supplementary figure 2 legend that the represented individual performed two spins. I strongly suggest the following edits to resolve this concern.

1) Mention in the text the finding of multiple rotations while diving (I would suggest providing as an average across all individuals) because without the supplementary information directly in front of the reader, the manuscript suggests otherwise. The term “spinning” dive is misleading if the dolphins only complete one revolution. By mentioning in the text (and suggestions for figures mentioned later) it will be apparent to the reader they spin multiple time.

2) I am curious if the individual that performed two rotations is an average number of spins. Figure S2 is representative of one individual so if the average number of spins across all animals is around one, to term the dive “spinning” is misleading. If you provide the number of spins and clarify in the text it will greatly support your naming of this new dive strategy.

3) To further clarify this concern, if you have video that shows a dolphin diving and completing a full rotation, (I am assuming unlikely since they are too deep to see from a drone at this point in the spin) it would be more representative of a “spinning” dive. From the video, it looks more like a roll.

Methods

84 – You state the data comes from 7 individuals; I would suggest clarifying if you used one dive sequence (descent and subsequent ascent) per animal or otherwise. Later in the results it states 8 recordings from 7 individuals, I suggest clarifying if one individual was tagged twice or if you used two separate recordings from the same tag.

87 – “Tags were generally...” – if there was an individual in the study that had its tag in a different location, I suggest it be addressed in the methods. If not, I do not think it’s necessary to say “generally”.

117 – Consider rewording for sentence flow, the end of the sentence is unclear.

119 – I am confused if the low-pass and high-pass filter was applied in on an additional software or in Matlab. If other software was used, you should expand the methods to include it. Otherwise, consider rewording for clarification.

123 – I think the methods section would be strengthened if you clarify why the filter-cut off frequency was set at 70%

Results

217 – Consider combining “This spin is...” with “This is shown” to better flow.

Figures

Figure 1 – For consistency reasons and to help with the comparison between dive types, I believe it would be easier to interpret if parts a,b,h,i were to be made into a separate figure. I suggest keeping the graphical representation (b and i) but using multiple stills to replace parts a and h. The single photograph used in part a does not contribute a lot of information about the spin behavior, however if you used several stills in a sequence above the drawing (b), it would be

concise and clear to the reader. Doing the same for the non-spinning dive next to it would allow for easier comparison.

I believe the latter part of figure 1 (parts c-g and j-n) as presented can then be replaced by Figure S2 (rename it figure 2) because it shows a longer time scale and still represents your trends from figure 1 very well. Additionally, by making these graphs their own figure, you can also make them larger and easier to interpret.

Figure 2 – I think this figure is well made and should be made figure 3.

Figure 3 – I think this can be moved to supplementary material considering it is referenced once in the results and discussion section. Also, it can easily be described verbally by the text unlike your other figures that I believe are more visually representative of the results.

Table 1 – One individual did not perform any spinning dives and this respective animal was tagged just before 2pm local time and the tag was deployed for 11 hours. This is my pure curiosity and interest, but it would be satisfying as a reader if you presented any life history information (if you have any) in Table 1.

Video S1 – I think it would be beneficial to make the video a similar, side by side comparison like parts a and b of figure one. I suggest playing video side by side of a spin vs non spinning dive and insert the graphical representation in the corner/on the side to help readers visualize the movements. I personally do not think the half speed and associated text are necessary.

Overall, I recommend acceptance of this manuscript with major edits. I hope the authors will take my suggestions to further strengthen their publication. I am extremely excited about this manuscript; I think it will be greatly beneficial to the field of marine mammal diving behavior and ecology as it fills a gap in knowledge of marine predator foraging in the pelagic ocean.

===PREPARING YOUR MANUSCRIPT===

If you have been asked to revise the written English in your submission as a condition of publication, you must do so, and you are expected to provide evidence that you have received language editing support. The journal would prefer that you use a professional language editing

service and provide a certificate of editing, but a signed letter from a colleague who is a native speaker of English is acceptable. Note the journal has arranged a number of discounts for authors using professional language editing services (<https://royalsociety.org/journals/authors/benefits/language-editing/>).

===PREPARING YOUR REVISION IN SCHOLARONE===

<https://royalsociety.org/journals/authors/author-guidelines/#supplementary-material> to include a suitable title and informative caption. An example of appropriate titling and captioning

may be found at https://figshare.com/articles/Table_S2_from_Is_there_a_trade-off_between_peak_performance_and_performance_breadth_across_temperatures_for_aerobic_sc_ope_in_teleost_fishes_/3843624.

Author's Response to Decision Letter for (RSOS-202320.R0)

See Appendix B.

RSOS-202320.R1 (Revision)

Review form: Reviewer 2

Is the manuscript scientifically sound in its present form?

Yes

Are the interpretations and conclusions justified by the results?

Yes

Is the language acceptable?

Yes

Do you have any ethical concerns with this paper?

No

Have you any concerns about statistical analyses in this paper?

No

Recommendation?

Accept with minor revision (please list in comments)

Comments to the Author(s)

See attached Comments to the Authors (Appendix C).

Review form: Reviewer 3

Is the manuscript scientifically sound in its present form?

Yes

Are the interpretations and conclusions justified by the results?

Yes

Is the language acceptable?

Yes

Do you have any ethical concerns with this paper?

No

Have you any concerns about statistical analyses in this paper?

No

Recommendation?

Accept with minor revision (please list in comments)

Comments to the Author(s)

The authors have made appropriate edits to the reviewers suggestions. Regarding my specific suggestions, sentences that I felt required rewording or revision for flow were corrected and clarification in the methods and about life history of study animals was made where necessary. Additionally, I am content with the decision to change the name of the dive to "spin" dive rather than "spinning" considering the potential confusion it may cause to readers. The method of selecting the new term by asking native speakers I feel was a justified means of deciding the new name. I have very minor comments and additional suggestions for edits as follows.

L65 - "Their target foraging depth, between around 50-600 m..." I suggest either removing "between" from this sentence or changing to "between ~50-600 m" for clarity.

L94 - Great description of active surface behavior to describe the onset of the spin dive.

L104-105 - Reword, you have "using" and then "used" in the same sentence which is redundant.

L131 - This sentence ends in "...on the tag following". Based on your response to another reviewer's suggestion to revise this sentence previously, I believe you are saying you followed methods from reference 17 to compute pitch and lateral rotation? If so, I believe it would be more clear to name the reference or add "methods used in previous tag studies" with the reference number at the end.

L160 - The description for defining forward speed was a great addition.

L182 - Spell out "August"

L200-202 - The description for variation in lateral rotation clarifies a lot to the reader and I think is a great addition to the paragraph.

L228-242 - This paragraph was an excellent addition and strengthens the manuscript.

L251-260 - I believe another reviewer previously suggested the inclusion of aerobic dive limit calculations and I agree it is an interesting point to include in this manuscript. Nicely done!

L335 - I believe using "preference" rather than "prevalence" would be more appropriate for this sentence.

Figure 2 - I thoroughly enjoyed the revisions made to figure 2. While I thought it was excellent in the first review, the edits to part c are superb. Excellent figure!

Figure 3 - I'm glad you added the R squared value to the description.

Table 2 - I support your decision to add foraging time in dive (%) to this table, I think that was a good choice.

Supplementary material

Figure S1 - I would spell out "four" rather than use "4 dive types".

Supplementary video - I know you have the permit listed in the upper right hand corner of the video but it may be a good idea to list the permit number in the video description in the supplementary download document as well.

Supporting Data CSV file download - My only recommendation would be to change the column headings to include units for all measurements, for example you labeled "Dive Time_s" so rename the others similarly.

I believe this manuscript is an excellent addition to the marine mammal literature and provides strong support for a knowledge gap in cetacean diving behavior. I recommend the editor to accept this manuscript following the minor revisions I've listed above.

Decision letter (RSOS-202320.R1)

Dear Dr Visser

On behalf of the Editors, we are pleased to inform you that your Manuscript RSOS-202320.R1 "Risso's dolphins perform spin dives to target deep-dwelling prey" has been accepted for publication in Royal Society Open Science subject to minor revision in accordance with the referees' reports. Please find the referees' comments along with any feedback from the Editors below my signature.

Please submit your revised manuscript and required files (see below) no later than 7 days from today's (ie 26-Oct-2021) date. Note: the ScholarOne system will 'lock' if submission of the revision is attempted 7 or more days after the deadline. If you do not think you will be able to meet this deadline please contact the editorial office immediately.

on behalf of Pete Smith (Subject Editor)
 openscience@royalsociety.org

Associate Editor Comments to Author:
 Comments to the Author:

With further apologies for the unusual delay in completing the review of your paper, thank you for so closely engaging with the queries of the reviewers - other than a number of largely typographical modifications, the reviewers are satisfied by the changes made and recommend your paper be published once you've taken these changes into account and incorporated them into your paper. Good luck and we'll look forward to receiving the final version of the paper in due course.

Reviewer comments to Author:
 Reviewer: 2
 Comments to the Author(s)
 See attached Comments to the Authors

Reviewer: 3
 Comments to the Author(s)

The authors have made appropriate edits to the reviewers suggestions. Regarding my specific suggestions, sentences that I felt required rewording or revision for flow were corrected and clarification in the methods and about life history of study animals was made where necessary. Additionally, I am content with the decision to change the name of the dive to "spin" dive rather than "spinning" considering the potential confusion it may cause to readers. The method of selecting the new term by asking native speakers I feel was a justified means of deciding the new name. I have very minor comments and additional suggestions for edits as follows.

L65 - "Their target foraging depth, between around 50-600 m..." I suggest either removing "between" from this sentence or changing to "between ~50-600 m" for clarity.

L94 - Great description of active surface behavior to describe the onset of the spin dive.

L104-105 - Reword, you have "using" and then "used" in the same sentence which is redundant.

L131 - This sentence ends in "...on the tag following". Based on your response to another reviewer's suggestion to revise this sentence previously, I believe you are saying you followed methods from reference 17 to compute pitch and lateral rotation? If so, I believe it would be more clear to name the reference or add "methods used in previous tag studies" with the reference number at the end.

L160 - The description for defining forward speed was a great addition.

L182 - Spell out "August"

L200-202 - The description for variation in lateral rotation clarifies a lot to the reader and I think is a great addition to the paragraph.

L228-242 - This paragraph was an excellent addition and strengthens the manuscript.

L251-260 - I believe another reviewer previously suggested the inclusion of aerobic dive limit calculations and I agree it is an interesting point to include in this manuscript. Nicely done!

L335 - I believe using "preference" rather than "prevalence" would be more appropriate for this sentence.

Figure 2 - I thoroughly enjoyed the revisions made to figure 2. While I thought it was excellent in the first review, the edits to part c are superb. Excellent figure!

Figure 3 - I'm glad you added the R squared value to the description.

Table 2 - I support your decision to add foraging time in dive (%) to this table, I think that was a good choice.

Supplementary material

Figure S1 - I would spell out "four" rather than use "4 dive types".

Supplementary video - I know you have the permit listed in the upper right hand corner of the video but it may be a good idea to list the permit number in the video description in the supplementary download document as well.

Supporting Data CSV file download - My only recommendation would be to change the column headings to include units for all measurements, for example you labeled "Dive Time_s" so rename the others similarly.

I believe this manuscript is an excellent addition to the marine mammal literature and provides strong support for a knowledge gap in cetacean diving behavior. I recommend the editor to accept this manuscript following the minor revisions I've listed above.

===PREPARING YOUR MANUSCRIPT===

one version should clearly identify all the changes that have been made (for instance, in coloured highlight, in bold text, or tracked changes);

===PREPARING YOUR REVISION IN SCHOLARONE===

-- Ensure that your data access statement meets the requirements at <https://royalsociety.org/journals/authors/author-guidelines/#data>. You should ensure that you cite the dataset in your reference list. If you have deposited data etc in the Dryad repository, please only include the 'For publication' link at this stage. You should remove the 'For review' link.

-- If you are requesting an article processing charge waiver, you must select the relevant waiver option (if requesting a discretionary waiver, the form should have been uploaded, see 'File upload' above).

-- If you have uploaded any electronic supplementary (ESM) files, please ensure you follow the guidance at <https://royalsociety.org/journals/authors/author-guidelines/#supplementary-material> to include a suitable title and informative caption. An example of appropriate titling and captioning may be found at https://figshare.com/articles/Table_S2_from_Is_there_a_trade-off_between_peak_performance_and_performance_breadth_across_temperatures_for_aerobic_scope_in_teleost_fishes_/3843624.

Author's Response to Decision Letter for (RSOS-202320.R1)

See Appendix D.

Decision letter (RSOS-202320.R2)

Dear Dr Visser,

I am pleased to inform you that your manuscript entitled "Risso's dolphins perform spin dives to target deep-dwelling prey" is now accepted for publication in Royal Society Open Science.

The proof of your paper will be available for review using the Royal Society online proofing system and you will receive details of how to access this in the near future from our production

office (openscience_proofs@royalsociety.org). We aim to maintain rapid times to publication after acceptance of your manuscript and we would ask you to please contact both the production office and editorial office if you are likely to be away from e-mail contact to minimise delays to publication. If you are going to be away, please nominate a co-author (if available) to manage the proofing process, and ensure they are copied into your email to the journal.

on behalf of Prof Pete Smith (Subject Editor)
openscience@royalsociety.org

Appendix A

Dear Editors,

This study addresses a deep-diving strategy of a Risso's dolphin population off of Terciera Island, Azores, that is considered unique among cetaceans. Authors detected a diel shift in spinning vs. non-spinning dives in apparent response to vertical movements of the deep scattering layer. They also found that spinning dives were significantly deeper and faster than non-spinning dives, enabling the animals to reach abundant mesopelagic prey and ultimately gain a sufficient energetic reward to offset energetic losses during these deep dives.

This is a thorough and cogent paper which presents highly novel data that will contribute greatly to the field of cetacean foraging ecology, and pelagic feeding ecology in general. My comments are mostly minor, and call for clarifications and more information, especially in the first half of the manuscript. The statistical analyses are detailed and well justified. The authors draw appropriate conclusions from their data and their resulting interpretations and hypotheses appear sound.

Abstract

Lines 24-25: Does the literature suggest that deep-dwelling prey is more energetically rewarding than prey found in epipelagic waters? Be more specific here or restructure this sentence for accuracy.

Lines 26-28: Combine these two sentences into, "Dives started with intense stroking and right-sided lateral rotation, resulting in rapid descent and a potential minimization of energetic costs during foraging."

Line 30: Is there a depth cut-off for deep-dwelling prey? I think this is important for a testable hypothesis.

Lines 30-32: Rewrite this sentence into "Hunting depth traced the diel movement of the deep scattering layer, a dense aggregation of prey that resides deep during the day and near-surface at night."

Background

Line 53: Add a hyphen between "energy" and "conserving"

Lines 62-65: Give some information on the range of dive depths previously documented in the literature. This is important information to inform your hypothesis.

Methods

Line 85: "Version 3" is listed in an add location. Is this the third version of a particular company/brand? Name the specific product with "V.3" after.

Lines 87-88: Were tags placed in other locations outside of the dorsal fin to blowhole area? Where were those locations?

Lines 89-91: Give a bit more info for the reader here on the specific visual cues associated with this dive type.

Line 111: Insert the word "those" between "that" and Risso's." This specifies that you are investigating a single population (i.e. in one geographical location) of this animal.

Line 118: following...? Is there a word missing here?

Lines 127-128: Add a citation for the information that dives below 20m are “breathing dives.”

Line 144: Change “electronical” to “electronic”

Line 170: Change “records” to “recordings”

Line 170: In the beginning of the Methods section you state that fieldwork was conducted between 2012-2018 but here you reference work being done during 2019.

Results and Discussion

Line 179: Why take 8 recordings where there were only 7 individuals? How did you account for this replication of one of your animals in your analysis? Give more information and justification here.

Line 183: Rewrite into “At dive-onset in spinning dives,...”

Line 187: Add a hyphen between “dive” and “onset”

Line 221: Add a hyphen between “energetically” and “costly”

Lines 297-300: How are the statements similar, when the first sentence compares day vs. dusk and the second sentence compares dusk vs. night-time?

Lines 310-311: Include citations to backup this statement. This is the first time I am seeing this specific information (unless I missed it) so establish appropriate citations here or earlier in the results and discussion

Figures

Lines 483: Include the p-values and R^2 values for the figure 3 linear regressions, either in the figure legend or on the graph itself.

Appendix B

Response to Editor and Reviewers for manuscript: RSOS-202320

NEW Title: Risso's dolphins perform spin dives to target deep-dwelling prey

Associate Editor Comments to Author:

Please accept our apologies for the unusual delay in completing the review of your paper - as we're sure you can imagine, it has proved exceptionally hard in recent months to secure the support of reviewers for many of the papers handled by the editors. We are, therefore, extremely grateful to the three commentators who have provided such extensive feedback on your paper. While it seems your work is broadly on track for acceptance, there are a number of modifications recommended by the reviewers that we'd like you to make - hopefully they won't be too onerous to enact, but we'd like to give you sufficient time to make the changes (hence the 3-week deadline). As one of the reviewers comments that their recommendations are major, we will ask for their advice after receipt of your revision, but we hope this will be a quick turnaround. Thanks again for your support.

Dear Editor. Thank you for the positive evaluation of our manuscript, and we entirely understand the challenges that these times bring. We were happy to read the very positive and constructive feedback provided by the three reviewers. We also much appreciate your understanding in extending the deadline for resubmission.

We have now completed the revision of our manuscript, following the reviewers' comments, including detailed alterations and response to the comments of Reviewer nr 3. In particular, we have added a new paragraph detailing the dive phases and have added several dive metrics. Moreover, after consultation with several native speakers, we have changed the name of the new dive type from 'spinning dive' to 'spin dive'. We provide a point-by-point response to the comments below (line numbers refer to the annotated version of the manuscript).

Reviewer 1 Comments:

This is a nice and well-written story that I greatly enjoyed reading. However, I do believe that this story can be completed by some extra analyses and a bit of theoretical background regarding the optimal diving theory to be made stronger. You will find below my main comments and suggestions.

1) L207: not sure about what you mean by greater forward speed (there is the absolute swimming speed and the vertical speed).

So is the greater vertical speed in spinning dives greater is due to a steeper pitch (descent angle) and a greater swimming speed or just a greater descent pitch? So is the initial acceleration aiming at reaching a greater absolute swimming speed when the Risso dolphins are gliding down on the spinning dives?

Thank you for this comment. We have now defined forward speed more clearly, as the speed in the direction of motion of the animal. It was calculated from the vertical speed (change in depth over a certain time interval), corrected for the angle of orientation (pitch) of the animal. The forward speed calculation (termed 'orientation-corrected depth rate', or ocd_r) was developed by Miller et al. 2004, and we have now included reference to this paper. Our calculation of forward speed is now more clearly explained on lines 160-164 of the annotated manuscript.

Our speed calculation was thus corrected for pitch and the observed greater speed in spin dives is independent of pitch. Spin dives both have a greater forward speed (and thereby a greater swimming speed), and a steeper pitch – joint adaptations aiming to get deeper faster.

2) L210: You never mention in your paper the Aerobic Dive Limit (or the Behavioral ADL). Could you estimate what is the difference in diving expenditure and on ADL between the two types of deep dives. Are Risso exceeding their bADL when performing spinning dive. Just lot the surface duration consecutive to the dive duration, you could also include an index of overall swimming effort through the dive.

This indeed is an interesting point that we have looked into in some detail. The aerobic dive limit (cADL) has been previously assessed for Risso's dolphins in the Pacific, performing comparable dive types (Arranz et al. 2019). They found a cADL ranging between 14.8 and 16.2 min and 8.9–9.7 min for adult and non-adult Risso's dolphins, respectively (excluding the spin at dive-onset). Thus, both spin and non-spin dives would be within the estimated cADL for this species, and individuals foraged mainly aerobically. This has now been added to the manuscript (L. 252-257).

As also indicated by Arranz et al., these ADL numbers are estimates, and carry some uncertainties (arising for example from parameter estimates in these equations originating from data from different cetacean species, since exact numbers for free ranging cetacean species are generally difficult to obtain). A similar matter would arise in the calculation of diving energetic expenditure as a function of dive type: it is difficult to move beyond relative best estimates – also as its interpretation is intricately linked to energetic gain from prey.

We therefore decided to include the direct comparison of energetic expenditure at dive onset (the discrimination on which the paper is focussed), to characterise the difference in effort between dive-onset in spin vs. non-spin dives, which is significantly larger in the first (MSA; Table 2), and provide a more extensive account on energetic expenditure and gain to our follow-up manuscript. This manuscript details energetic expenditure as a function of dive type, and in relation to energetic intake from prey, using an energetic modelling approach.

Does the relationship between dive duration and the post dive interval varies (or not) between spinning and non-spinning dives? This should nice to look at.

Interesting point. We inspected the relationship and they are comparable between the two dive types, with a mode of dive interval = 1.3 x dive duration for both types. This is now detailed in Lines 254-256.

3) L229: This is the information I was missing§ I was wondering if the Risso Dolphins were actively stroking through the whole dive or not. So is the idea is to increase the initial speed as

much as possible prior to the gliding phase to descend as quickly as possible, but this swimming effort should last until the Risso's are fully compressed and become negatively buoyant (to glide down). Maybe that what is the biggest constraint on these animals is the amount of oxygen stores and therefore this is the best strategy allowing them to spend more time at depth for a small increase of energy expenditure at the beginning of the dive, I guess this could be modelled. See also the work performed on pilot whales in the Strait of Gibraltar.

I would encourage you to describe more precisely the different phases of the dive, we can see from your figure 1 that the Risso's stroke for approximately 20 seconds at the beginning of the dive and then they glide. However, this does not tell us the depth at which they are nearly fully compressed and become negatively buoyant to glide down.

So I wonder if the diving strategy in spinning dives is to initiate the glide with a greater initial speed to go down as quickly as possible, you have all the elements to verify that and to state it clearly if it is the case. In my view the dive strategy could be described more precisely and the whole story will be easier to follow, mostly for people who are not familiar with diving behavior studies.

Very good suggestion, thank you. We have now added a new paragraph detailing the dive phases (L. 282-242) and have further clarified throughout the manuscript. Risso's dolphins actively stroke during the first part of their foraging dive descent, then glide down and start active stroking again at onset of active foraging (Figs. 1, S2). This becomes apparent from the example in Fig. 1d and k, showing stroking up to 21 s (d, spin dive) and 14 s (k, non-spin dive) into the descent, followed by gliding (L. 543-544). In this example, gliding starts at a depth of respectively 40 m (spin dive) and 20 m (non-spin dive). Indeed, the aim is to increase speed and depth as much as possible before initiating the gliding phase (L. 240-242, 265-268). Moreover, individuals exhale at onset of the spin dive, suggesting they release air volume to further reduce buoyancy and thus enable larger speed at glide onset (L. 238-240).

4) L239: I do believe that you have all the information necessary to investigate if Risso's are targeting different prey items between day and night and during the day between spinning dives and non-spinning ones. Have you been looking at the echoes of the clicks that you should be able to get on the D-tag, do they tell you something about possible differences in targeted prey's sizes between the two dive categories with the hypothesis that when performing spinning dives Risso's might be targeting more rewarding prey (i.e. larger preys) compared to non-spinning dives. Just a guess. Does the chase behavior in terms of swimming effort, direction changes of the prey is identical or differ between those two situations. If I understood well what you suggest, during the day preys are deeper and in colder waters and as they are ectotherms, they are likely to be less active compared to the night when preys are closer to the surface in warmer water (what is the temperature gradient).

Good suggestion. We would indeed be very interested to look into the return echoes from prey. For Risso's dolphin, however, these are not recorded on the tag, likely due to physical properties of the melon (i.e. echoes are blocked by melon) (M. Johnson, unpublished data). So we are not able to assess prey characteristics such as size or escape behaviours and speed from the tag data. Comparable foraging rates and inter-capture-intervals between spin and non-spin bouts indicates

foraging on possibly similar prey, optimising prey capture rate vs. depth as a function of prey behaviour and catchability (L. 318-326). It could be that this also relates to being able to target larger, or otherwise more calorific prey in one dive type over the other. We cannot assess that from our data, but the comparable foraging rate does not suggest a major difference in overall energetic reward per prey item (L. 323-326).

5) L255 change in the pitch of the descent in relation to the foraging success of the previous dive has been found in many diving seabirds and marine mammals equipped with accelerometers and pressure sensor. Most of the time they increase their dive angle to return more quickly at depth and maximize the bottom duration of their dive (the main foraging phase of the dive). You neither refer to the bottom duration of the dive and I would encourage you to do so to verify how this change according to diving depth and between spinning versus non spinning dives.

Risso's dolphin dives are variable in shape and do not have an easily discernible bottom phase, and individuals forage also during descent and ascent. Therefore, we defined the foraging period in the dive (comparable to bottom phase defined for other species), as the period from the first to the last buzz (% foraging time; L. 155-157). The steep, faster descent of spinning dives indeed results in a slightly longer relative foraging period for spinning dives (49%), in relation to non-spinning dives (42%). This is now integrated in the manuscript (L. 234-236, and table 2). In this manuscript we focus on the discrimination between spin and non-spin dives. Dive-by-dive variation in relation to foraging return is indeed of interest and part of the previously mentioned follow-up energetic modelling effort (see our response to point 2).

6) L277 ok but this should be expressed per unit of time spent diving + recovering (not per dive) as indicated above differences in targeted prey sizes might be part of the equation and should be discussed.

We agree, and this is actually how it is described: rate of prey capture attempts within foraging bouts (consecutive dives + inter dive intervals; L. 170-174). We have further clarified the wording in the manuscript (L. 318). As discussed in point 4, prey size could indeed be part of the equation. However, the fact the individuals show comparable foraging rates and forage in the same layer (but at different depths) between dive types/bouts, indicates comparable prey. We have grounds to expect variation in catchability of individuals through changes in the environment (water temperature), schooling (density, group defence) and behaviour (refuge vs. foraging) between the foraging depths. Thus, under some circumstances it pays off to dive deeper for comparable prey (because more can be caught at lesser expense). Prey size adds a further (but in our case unknown) dimension to this equation, that can shift the balance one way or the other. If much larger prey was targeted in one dive type over the other, this would likely relate to a (strong) reduction in buzz rates for deeper dives, as is found in pilot whales (many vs. 1 or 2 buzzes in shallower vs. deep dives; Aguilar de Soto et al. 2008). Instead, we find the opposite, doubling of per-dive buzz rate for deeper dives and comparable buzz rates within foraging bouts. This is now more prominently discussed in the manuscript (L. 323-326).

Reviewer 2 Comments:

Dear Editors,

This study addresses a deep-diving strategy of a Risso's dolphin population off of Terciera Island, Azores, that is considered unique among cetaceans. Authors detected a diel shift in spinning vs. non-spinning dives in apparent response to vertical movements of the deep scattering layer. They also found that spinning dives were significantly deeper and faster than non-spinning dives, enabling the animals to reach abundant mesopelagic prey and ultimately gain a sufficient energetic reward to offset energetic losses during these deep dives.

This is a thorough and cogent paper which presents highly novel data that will contribute greatly to the field of cetacean foraging ecology, and pelagic feeding ecology in general. My comments are mostly minor, and call for clarifications and more information, especially in the first half of the manuscript. The statistical analyses are detailed and well justified. The authors draw appropriate conclusions from their data and their resulting interpretations and hypotheses appear sound.

Thank you for your very positive evaluation!

Abstract

1) Lines 24-25: Does the literature suggest that deep-dwelling prey is more energetically rewarding than prey found in epipelagic waters? Be more specific here or restructure this sentence for accuracy.

We have changed this sentence to state "Cetacean optimal foraging entails a tight balance between oxygen-conserving dive strategies and access to energetically rewarding deep-dwelling prey of sufficient energetic reward." (L. 24-26).

2) Lines 26-28: Combine these two sentences into, "Dives started with intense stroking and right-sided lateral rotation, resulting in rapid descent and a potential minimization of energetic costs during foraging."

We have changed wording to: "Dives started with intense stroking and right-sided lateral rotation. This remarkable behaviour resulted in rapid descent." (L. 27-29).

3) Line 30: Is there a depth cut-off for deep-dwelling prey? I think this is important for a testable hypothesis.

We hypothesized spinning dives to be 1) foraging dives and 2) targeting deep layers. The actual discrimination of target depth between dive types was a result of our analysis. We did not have an a priori expectation (or depth cut-off) of actual depth, or of the strict discrimination that was found between spin and non-spin dives in terms of target depth.

4) Lines 30-32: Rewrite this sentence into "Hunting depth traced the diel movement of the deep scattering layer, a dense aggregation of prey that resides deep during the day and near-surface at night.

Done (L. 33).

Background

5) Line 53: Add a hyphen between “energy” and “conserving”

Done (L. 54).

6) Lines 62-65: Give some information on the range of dive depths previously documented in the literature. This is important information to inform your hypothesis.

Done, we have added that individuals forage between 50-600 m deep (L. 65).

Methods

7) Line 85: “Version 3” is listed in an add location. Is this the third version of a particular company/brand? Name the specific product with “V.3” after.

Done (L. 88).

8) Lines 87-88: Were tags placed in other locations outside of the dorsal fin to blowhole area? Where were those locations?

Either dorsally or on the flank. This has been added to the description (L. 90-91).

9) Lines 89-91: Give a bit more info for the reader here on the specific visual cues associated with this dive type.

Done, we have added that “spin is an active surface behaviour (near-surface acceleration plus rotation, inducing a marked trail of white water) that can be reliably characterised from visual observations” (L. 93-94).

10) Line 111: Insert the word “those” between “that” and Risso’s.” This specifies that you are investigating a single population (i.e. in one geographical location) of this animal.

We have added reference to the geographical location (Azores) (L. 116).

11) Line 118: following...? Is there a word missing here?

Following methods described in reference number 17 (L. 131).

12) Lines 127-128: Add a citation for the information that dives below 20m are “breathing dives.”

Done (L. 124).

13) Line 144: Change “electronical” to “electronic”

Done (L. 150).

14) Line 170: Change “records” to “recordings”

Done (L. 179).

15) Line 170: In the beginning of the Methods section you state that fieldwork was conducted between 2012-2018 but here you reference work being done during 2019.

Thank you for spotting this mistake, we have changed it to 2012-2019 (L. 86).

Results and Discussion

16) Line 179: Why take 8 recordings where there were only 7 individuals? How did you account for this replication of one of your animals in your analysis? Give more information and justification here.

We accounted for this replication by using animal ID (and not tag ID) in our statistical analysis. This is stated in the description of our statistical analysis in the Methods Section (L. 176).

17) Line 183: Rewrite into “At dive-onset in spinning dives,…”

Sentence was rewritten (L. 195).

18) Line 187: Add a hyphen between “dive” and “onset”

Done (L. 203).

19) Line 221: Add a hyphen between “energetically” and “costly”

Done (L. 261).

20) Lines 297-300: How are the statements similar, when the first sentence compares day vs. dusk and the second sentence compares dusk vs. night-time?

Spin dives are day time dives mostly, and non-spinning dives are night-time dives mostly, with the transition of spin to non-spin occurring around dusk. This generates a daily sequence of: 1) day time spin – 2) dusk spin – 3) dusk non-spin – 4) night-time non-spin dives. We indicate that within both spin and non-spin dives, those around dusk have a higher buzz rate than the ones of the same type occurring either before (day time) or after (night time) dusk. This is now further clarified (L. 341-346).

21) Lines 310-311: Include citations to backup this statement. This is the first time I am seeing this specific information (unless I missed it) so establish appropriate citations here or earlier in the results and discussion

Done, we have added 3 citations for terrestrial and marine top predators (L. 357).

Figures

22) Lines 483: Include the p-values and R² values for the figure 3 linear regressions, either in the

figure legend or on the graph itself.

The values were added to the figure legend of figure 3 (L. 564-565).

Reviewer 3 Comments:

I feel this manuscript will contribute greatly to advancing scientific literature and increasing our knowledge regarding cetacean diving behavior and its role in marine ecosystems. To summarize the manuscript, the authors aimed to characterize a unique dive type (termed a spinning dive) used by Risso's dolphins of the Azores and interpret the function of this dive strategy. The authors hypothesized this newly described spinning dive is utilized to target deep-dwelling prey and to optimize foraging performance. The authors tested this hypothesis using tag data from seven Risso's dolphins and compared the behavior and kinematics between spinning and non-spinning dives. The main take-away from this study is that Risso's dolphins both proactively plan and then utilize a metabolically costly spinning dive to reach deep-dwelling prey, and while it poses an energetic risk, this dive strategy is sufficiently rewarded with access to a densely populated prey environment.

I believe the authors were thorough when justifying reasoning for their methods and that the statistics are sound. While reading, I found the writing was concise and clear (except for a minor few instances, see suggestions line by line). The structure is well-organized, and the manuscript is a good length. I believe the topic is remarkably interesting and the authors connected their findings into a gap in knowledge of cetacean diving strategies. While the manuscript is satisfactory, I feel there are changes that should occur prior to publication. I would like to address my suggestions of this manuscript, in order of each section, line-by-line.

Thank you very much for your thorough and positive review, and the description of our manuscript in relation to the broader field. We have responded to your comments point by point below and have changed our manuscript accordingly.

Background

1) 62 and 63 – Revise for flow, maybe consider combining the first two sentences.

Sentence has been revised for flow (L. 64-65).

2) 70 – When I hear the word spinning, I think of a spinner dolphin or spinner shark, both of which make several 360 revolutions while spinning. From Figure 1 and the supplementary video, it appears the dolphins may only complete one spin. Additionally, it is not mentioned in the text if they spin multiple times. I found in the supplementary figure 2 legend that the represented individual performed two spins. I strongly suggest the following edits to resolve this concern.

1) Mention in the text the finding of multiple rotations while diving (I would suggest providing as an average across all individuals) because without the supplementary information directly in front of the reader, the manuscript suggests otherwise. The term "spinning" dive is misleading if the dolphins only complete one revolution. By mentioning in the text (and suggestions for figures mentioned later) it will be apparent to the reader they spin multiple time. 2) I am curious if the individual that performed two rotations is an average number of spins. Figure S2 is representative of one individual so if the average number of spins across all animals is around one, to term the

dive "spinning" is misleading. If you provide the number of spins and clarify in the text it will greatly support your naming of this new dive strategy.

3) To further clarify this concern, if you have video that shows a dolphin diving and completing a full rotation, (I am assuming unlikely since they are too deep to see from a drone at this point in the spin) it would be more representative of a "spinning" dive. From the video, it looks more like a roll.

Thank you for this comment. We aimed to specifically refer to the performance of a body rotation at dive-onset, associated with a strong acceleration and steep pitch (described in L. 195-200). In the Results section we state the degrees of rotation during the first second at dive-onset, associated to this acceleration (63-171 degrees). This behaviour enables maximisation of speed when entering the gliding phase (L. 240-242). Individuals then typically, but not always, completed a full rotation into the descent (as exemplified in Fig. S2; L. 200-202). We agree that the name 'spinning dive' in this context could potentially be confusing if understood as the individual performing multiple spins. In order to select the best possible term, we have described the dive-onset (strong acceleration + a partial or full body rotation) to several native speakers and requested their advice on the name. We provided several options, including 'twist dive', 'spin dive', 'rotation dive', 'sprint dive' and 'corkscrew dive', or 'other suggestion'. The consensus from this consultation round was that the best term to use is 'a spin dive'. This term describes a fast rotation, that can, but not necessarily needs to be a full body rotation (i.e. the special behaviour we describe). A 'spin dive' also does not suggest multiple spins, whereas this would have been the case for a 'spinning dive'.

We have therefore changed the name to 'spin dive' throughout.

Individuals could perform multiple full rotations during descent, but this was variable and not restricted to spin dives (i.e., a general characteristic of both spin and non-spin descent gliding-phases). Individuals performed between 0 and 3 rotations during descent in spin and non-spin dives, excluding the spin at dive-onset (mean (SD) 0.9 (1.0)). The two rotations in the spin dive example in Fig. S2 is thus close to average. This rotation behaviour during gliding is not specific to spin dives and not included in the naming of the dive type.

Based on your comments and on the comments of Reviewer 1, we have now more clearly described the dive phases and rotation behaviour in the manuscript (L. 228-242). We unfortunately cannot track dolphins deeper using video recordings than in the video we provide.

Methods

3) 84 – You state the data comes from 7 individuals; I would suggest clarifying if you used one dive sequence (descent and subsequent ascent) per animal or otherwise. Later in the results it states 8 recordings from 7 individuals, I suggest clarifying if one individual was tagged twice or if you used two separate recordings from the same tag.

The data originates from 8 tags deployed on 7 different individuals. From these 8 tag records, all foraging dives were analysed (not one dive sequence per tag record). This is specified in L. 120-123, 154-155, 191-192, table 1. It is stated in table 1 that tag records gg13_238a and gg17_203a are on the same individual, individual 1 (tagged in 2013 and in 2017).

4) 87 – "Tags were generally..." – if there was an individual in the study that had its tag in a different location, I suggest it be addressed in the methods. If not, I do not think it's necessary to say "generally".

Generally has been removed (L. 90).

5) 117 – Consider rewording for sentence flow, the end of the sentence is unclear.

Reworded sentence for flow (L. 120-122).

6) 119 – I am confused if the low-pass and high-pass filter was applied in on an additional software or in Matlab. If other software was used, you should expand the methods to include it. Otherwise, consider rewording for clarification.

Reworded for clarification (L. 102). All was performed using Matlab.

7) 123 – I think the methods section would be strengthened if you clarify why the filter-cut off frequency was set at 70%

We now further clarify that in order to remove dynamic movement information (from stroking) from the accelerometer data, we apply a low-pass filter at 70% of the dominant stroke frequency (L. 136-138).

Results

8) 217 – Consider combining "This spin is..." with "This is shown" to better flow.

This sentence was removed.

Figures

9) Figure 1 – For consistency reasons and to help with the comparison between dive types, I believe it would be easier to interpret if parts a,b,h,i were to be made into a separate figure. I suggest keeping the graphical representation (b and i) but using multiple stills to replace parts a and h. The single photograph used in part a does not contribute a lot of information about the spin behavior, however if you used several stills in a sequence above the drawing (b), it would be concise and clear to the reader. Doing the same for the non-spinning dive next to it would allow for easier comparison. I believe the latter part of figure 1 (parts c-g and j-n) as presented can then be replaced by Figure S2 (rename it figure 2) because it shows a longer time scale and still represents your trends from figure 1 very well. Additionally, by making these graphs their own figure, you can also make them larger and easier to interpret.

Thank you for your suggestions. We have carefully considered these alterations. For the first part, extending the photographic sequence, is something we have done in an earlier version of the graph and it did not clarify the sequence better than the current photograph + drawn sequence. The photographs show active movement, but because of the water movement and the individual

being partly under water, the sequence in photos does not show well what occurs. That is why we added the graphic sequence. The photos now indicate the difference in near surface orientation and energetic level at one point during dive-onset. The graphics indicate the phases. The two types of information (graphic + tag data) are combined in one graph to enable easier interpretation of the movement data, also for people not familiar with reading this kind of information: the graphics are timed to and represent the tag movement and orientation data. Given the emphasis in the dive-onset, we prefer to show a zoomed version that focusses on the dive-onset (current Fig. 1) over a zoomed-out version (current Fig. S2).

10) Figure 2 – I think this figure is well made and should be made figure 3.

Thank you!

11) Figure 3 – I think this can be moved to supplementary material considering it is referenced once in the results and discussion section. Also, it can easily be described verbally by the text unlike your other figures that I believe are more visually representative of the results.

Thank you for your suggestion. This has been discussed among the authors and we do feel it adds important visual information and data on foraging rates as a function of foraging bout, and bout type data that extends the description in the text.

12) Table 1 – One individual did not perform any spinning dives and this respective animal was tagged just before 2pm local time and the tag was deployed for 11 hours. This is my pure curiosity and interest, but it would be satisfying as a reader if you presented any life history information (if you have any) in Table 1.

It is difficult to distinguish between male and female adults in Risso's dolphins with certainty. We have clarified all individuals are adults (L. 87, table 1). The individual not performing spinning dives started foraging later in the evening, when the prey layer was already migrating to the surface.

Video S1 – I think it would be beneficial to make the video a similar, side by side comparison like parts a and b of figure one. I suggest playing video side by side of a spin vs non spinning dive and insert the graphical representation in the corner/on the side to help readers visualize the movements. I personally do not think the half speed and associated text are necessary.

Thank you for your suggestions. As the non-spinning dives are performed mainly at or after dusk, our ability to record them are strongly light-limited and we unfortunately have no comparable recording of a non-spinning dive.

Overall, I recommend acceptance of this manuscript with major edits. I hope the authors will take my suggestions to further strengthen their publication. I am extremely excited about this manuscript; I think it will be greatly beneficial to the field of marine mammal diving behavior and ecology as it fills a gap in knowledge of marine predator foraging in the pelagic ocean.

Thank you very much for your strong support of our manuscript and your in-depth review.

Appendix C

Journal: Royal Society Open Science

Manuscript ID: RSOS-202320.R1

Manuscript title: Risso's dolphins perform spin dives to target deep-dwelling prey

Comments to the authors:

The authors took great care in addressing the reviews from myself and the two other reviewers. Their responses were thorough and improved the clarification of the study, especially in the Methods section. This round, I detected a few content-related/several editorial changes needed, which I have outlined below.

Abstract

Line 25: Rather than using "unknown" to describe the Risso's dive, it may be more appropriate to use "uninvestigated" or "unstudied." "Unknown" implies a lack of anecdotes in addition to a lack of quantifiable data. Are there anecdotes that lead to the investigation of this unique diving pattern. This is more food for thought than a direct suggestion, just bear it in mind with your word selection here.

Background:

Line 59: Capitalize "i" at the beginning of the sentence.

Methods:

Line 82: Spell out 7, and do this for all numbers under ten that aren't followed by units (ex: meters).

Results and Discussion:

Line 182: Change 8 and 7 to word format.

Line 196: Change 6 and 7 to word format.

Line 199: Change 7 to word format.

Line 278: Add the word "as" between "appears" and "an."

Lines 209-301: It took me a little while to figure out which variable was being described here, I had to look up the values in table 2 to confirm. I would clear this up by changing the info presented within the parentheses into (number of buzzes; 10.7 vs. 5.5; table 2).

Lines 351-353: Reconsider the last sentence of your conclusion. You have shown that Risso's dolphins are intertwined with food webs as deep as mesopelagic prey, but the structure of your paper didn't include much background on other species of cetaceans foraging at the same/similar layers. Maybe rewrite into: "The proficient exploitation of pelagic deep-sea prey by Risso's dolphins demonstrates the potentiality of cetaceans to serve as key drivers that link deep and shallow ecologies and oceanic food web dynamics."

Figures:

Figure 2: Remind the reader that panels "a" and "b" are visual observation records. I was confused about why "a" data started at 10:00 and ended at 3:00 and why panel "b" data began at 6:00 and ended at 21:00. Clarify this further and even explain more in the Methods text.

Appendix D

Response to Editor and Reviewers for manuscript: RSOS-202320

Title: Risso's dolphins perform spin dives to target deep-dwelling prey

Associate Editor Comments to Author:

With further apologies for the unusual delay in completing the review of your paper, thank you for so closely engaging with the queries of the reviewers - other than a number of largely typographical modifications, the reviewers are satisfied by the changes made and recommend your paper be published once you've taken these changes into account and incorporated them into your paper. Good luck and we'll look forward to receiving the final version of the paper in due course.

Dear Editor. Thank you very much for this great news! We are very happy that our manuscript is accepted, pending on some final minor changes. We have processed the requested changes, outlined point-by-point below, and hereby resubmit the final version. (Line numbers in the response letter refer to the annotated version of the manuscript).

Reviewer 2 Comments:

The authors took great care in addressing the reviews from myself and the two other reviewers. Their responses were thorough and improved the clarification of the study, especially in the Methods section. This round, I detected a few content-related/several editorial changes needed, which I have outlined below.

We are very happy to hear our edits further improved and clarified the study, thank you for your detailed review of our manuscript.

Abstract

Line 25: Rather than using “unknown” to describe the Risso’s dive, it may be more appropriate to use “uninvestigated” or “unstudied.” “Unknown” implies a lack of anecdotes in addition to a lack of quantifiable data. Are there anecdotes that lead to the investigation of this unique diving pattern. This is more food for thought than a direct suggestion, just bear it in mind with your word selection here.

Thank you for the food for thought. Given the lack of anecdotes other than our own observations, we have decided to keep ‘unknown’ in this case.

Background:

Line 59: Capitalize “i” at the beginning of the sentence.

Done (L. 59).

Methods:

Line 82: Spell out 7, and do this for all numbers under ten that aren’t followed by units (ex: meters).

Done (L. 28, 82).

Results and Discussion:

Line 182: Change 8 and 7 to word format.

Done (L. 182).

Line 196: Change 6 and 7 to word format.

Done (L. 196).

Line 199: Change 7 to word format.

Done (L. 199).

Line 278: Add the word “as” between “appears” and “an.”

Included ‘to be’ to clarify wording (L. 279).

Lines 209-301: It took me a little while to figure out which variable was being described here, I had to look up the values in table 2 to confirm. I would clear this up by changing the info presented within the parentheses into (number of buzzes; 10.7 vs. 5.5; table 2).

Done (L. 300).

Lines 351-353: Reconsider the last sentence of your conclusion. You have shown that Risso’s dolphins are intertwined with food webs as deep as mesopelagic prey, but the structure of your paper didn’t include much background on other species of cetaceans foraging at the same/similar layers. Maybe rewrite into: “The proficient exploitation of pelagic deep-sea prey by Risso’s dolphins demonstrates the potentiality of cetaceans to serve as key drivers that link deep and shallow ecologies and oceanic food web dynamics.”

We have changed wording to: “Risso’s dolphin proficient exploitation of pelagic deep-sea prey illustrates the role of cetaceans as key drivers of deep and shallow oceanic food web dynamics and reveals direct ecological linkage between deep and shallow systems.” (L. 351-353).

Figures:

Figure 2: Remind the reader that panels “a” and “b” are visual observation records. I was confused about why “a” data started at 10:00 and ended at 3:00 and why panel “b” data began at 6:00 and ended at 21:00. Clarify this further and even explain more in the Methods text

Thank you for this suggestion, we have altered the legend of Figure 2 to clarify (L. 548-556).

Reviewer 3 Comments:

The authors have made appropriate edits to the reviewers suggestions. Regarding my specific suggestions, sentences that I felt required rewording or revision for flow were corrected and clarification in the methods and about life history of study animals was made where necessary. Additionally, I am content with the decision to change the name of the dive to "spin" dive rather than "spinning" considering the potential confusion it may cause to readers. The method of selecting the new term by asking native speakers I feel was a justified means of deciding the new name. I have very minor comments and additional suggestions for edits as follows.

Thank you very much for your detailed review and suggestions to our manuscript. We are happy to hear you were satisfied with our changes, and have made final edits based on your comments below.

L65 - "Their target foraging depth, between around 50-600 m..." I suggest either removing "between" from this sentence or changing to "between ~50-600 m" for clarity.

We have removed 'between' (L. 60).

L94 - Great description of active surface behavior to describe the onset of the spin dive.

Thank you!

L104-105 - Reword, you have "using" and then "used" in the same sentence which is redundant.

We changed 'using' to 'from' (L. 98).

L131 - This sentence ends in "...on the tag following". Based on your response to another reviewer's suggestion to revise this sentence previously, I believe you are saying you followed methods from reference 17 to compute pitch and lateral rotation? If so, I believe it would be more clear to name the reference or add "methods used in previous tag studies" with the reference number at the end.

We have added 'following standard methods for tag data analysis' (L. 125).

L160 - The description for defining forward speed was a great addition.

Thank you!

L182 - Spell out "August"

Done (L. 173).

L200-202 - The description for variation in lateral rotation clarifies a lot to the reader and I think is a great addition to the paragraph.

Thank you.

L228-242 - This paragraph was an excellent addition and strengthens the manuscript.

Thank you for your earlier suggestion! We agree this paragraph strengthens the manuscript.

L251-260 - I believe another reviewer previously suggested the inclusion of aerobic dive limit calculations and I agree it is an interesting point to include in this manuscript. Nicely done!

We agree, and thank you again for your positive input!

L335 - I believe using "preference" rather than "prevalence" would be more appropriate for this sentence.

Changed to preference (L. 324).

Figure 2 - I thoroughly enjoyed the revisions made to figure 2. While I thought it was excellent in the first review, the edits to part c are superb. Excellent figure!

Figure 3 - I'm glad you added the R squared value to the description.

Table 2 - I support your decision to add foraging time in dive (%) to this table, I think that was a good choice.

Happy to hear, thank you!

Supplementary material

Figure S1 - I would spell out "four" rather than use "4 dive types".

Done.

Supplementary video - I know you have the permit listed in the upper right hand corner of the video but it may be a good idea to list the permit number in the video description in the supplementary download document as well.

Done, we have added the permit number to the legend of Video S1.

Supporting Data CSV file download - My only recommendation would be to change the column headings to include units for all measurements, for example you labeled "Dive Time_s" so rename the others similarly.

Good suggestion, we have done so.

I believe this manuscript is an excellent addition to the marine mammal literature and provides strong support for a knowledge gap in cetacean diving behavior. I recommend the editor to accept this manuscript following the minor revisions I've listed above.

Thank you for your time and effort to review this manuscript in such a positive and constructive way. Your suggested edits were excellent and supported further strengthening of our manuscript.